EMBO
Molecular Medicine

# The diphenylpyrazole compound anle138b blocks Aβ channels and rescues disease phenotypes in a mouse model for amyloid pathology

Ana Martinez Hernandez[1,2,†], Hendrik Urbanke[1,†], Alan L Gillman[3,†] 🄳, Joon Lee[3,†], Sergey Ryazanov[4,5], Hope Y Agbemenyah[6], Eva Benito[1], Gaurav Jain[1], Lalit Kaurani[5], Gayane Grigorian[7], Andrei Leonov[4,5], Nasrollah Rezaei-Ghaleh[4,8], Petra Wilken[5,6,9], Fernando Teran Arce[3,‡], Jens Wagner[10], Martin Fuhrman[10], Mario Caruana[11], Angelique Camilleri[11], Neville Vassallo[11], Markus Zweckstetter[4,5,8,12], Roland Benz[13], Armin Giese[14], Anja Schneider[5,6,9], Martin Korte[7,15,*] 🄳, Ratnesh Lal[3,**] 🄳, Christian Griesinger[4,5,***] 🄳, Gregor Eichele[2,****] 🄳 & Andre Fischer[1,6,*****] 🄳

## Abstract

**Alzheimer's disease is a devastating neurodegenerative disease eventually leading to dementia. An effective treatment does not yet exist. Here we show that oral application of the compound anle138b restores hippocampal synaptic and transcriptional plasticity as well as spatial memory in a mouse model for Alzheimer's disease, when given orally before or after the onset of pathology. At the mechanistic level, we provide evidence that anle138b blocks the activity of conducting Aβ pores without changing the membrane embedded Aβ-oligomer structure. In conclusion, our data suggest that anle138b is a novel and promising compound to treat AD-related pathology that should be investigated further.**

**Keywords** Alzheimer's disease; amyloid pathology; Aβ channels; gene expression; membrane pores

**Subject Categories** Chromatin, Epigenetics, Genomics & Functional Genomics; Neuroscience; Pharmacology & Drug Discovery

See also: **FX Guix & CG Dotti** (January 2018)

## Introduction

Alzheimer's disease (AD) is the most common neurodegenerative disorder causing a severe emotional and economic burden to our societies. Due to increased life expectancies, the number of those afflicted with AD is expected to double by 2025. Despite intensive research, effective therapeutic approaches are still not available. The pathogenesis of AD has been linked to protein aggregation,

1   Department for Epigenetics and Systems Medicine in Neurodegenerative Diseases, German Center for Neurodegenerative Diseases (DZNE), Göttingen, Germany
2   Department for Genes and Behavior, Max Planck Institute for Biophysical Chemistry, Göttingen, Germany
3   Department of Bioengineering, Materials Science and Engineering, Department of Mechanical and Aerospace Engineering and Institute of Engineering in Medicine, University of California San Diego, La Jolla, CA, USA
4   Department of NMR Based Structural Biology, Max Planck Institute for Biophysical Chemistry, Göttingen, Germany
5   DFG Research Center Nanoscale Microscopy and Molecular Physiology of the Brain, Göttingen, Germany
6   Department of Psychiatry and Psychotherapy, University Medical Center Göttingen, Göttingen, Germany
7   Department of Cellular Neurobiology, Technical University Braunschweig, Braunschweig, Germany
8   Department of Translational Structural Biology of Dementia, German Center for Neurodegenerative Diseases (DZNE), Göttingen, Germany
9   Group for Translational Research in Neurodegenerative Diseases, German Center for Neurodegenerative Diseases (DZNE) Göttingen, Göttingen, Germany
10  Group for Neuroimmunology and Imaging, German Center for Neurodegenerative Diseases (DZNE), Bonn, Germany
11  Department of Physiology and Biochemistry, Centre for Molecular Medicine and Biobanking, University of Malta, Msida, Malta
12  Department of Neurology, University Medical Center Göttingen, University of Göttingen, Göttingen, Germany
13  Life Sciences and Chemistry, Jacobs University of Bremen, Bremen, Germany
14  Center for Neuropathology and Prion Research, Ludwig-Maximilians-University Munich, Munich, Germany
15  Helmholtz Center for Infections Research, Braunschweig, Germany
    ‡Present address:Departments of Medicine and Biomedical Engineering, University of Arizona, Tucson, AZ, USA
    *Corresponding author. Tel: +49 531 391 3220; E-mail: m.korte@tu-braunschweig.de
    **Corresponding author. Tel: +1 858 822 0384; E-mail: rlal@ucsd.edu
    ***Corresponding author. Tel: +49 551 201 2201; E-mail: cigr@nmr.mpibpc.mpg.de
    ****Corresponding author. Tel: +49 551 201 2701; E-mail: gregor.eichele@mpibpc.mpg.de
    *****Corresponding author. Tel: +49 551 3961211; E-mail: afische2@gwdg.de
    †These authors contributed equally to this work

namely the aggregation of amyloid-beta peptides (Aβ) and tau protein. The accumulation of pathogenic aggregates of Aβ peptides in the brain appears to be a key event in the pathogenesis of AD (Iversen *et al*, 1995; Tanzi, 2005; Jakob-Roetne & Jacobsen, 2009; Goate & Hardy, 2012), and targeting amyloid pathology still represents a promising therapeutic strategy (Haass & Selkoe, 2007; Sevigny *et al*, 2016). The precise molecular events that trigger amyloid-induced decline of synaptic plasticity and neuronal cell death are still not entirely resolved and are likely to be multifactorial. One of the first explanations of neuronal dysfunction and toxicity in AD is the channel hypothesis first proposed by Arispe and coworkers (Arispe *et al*, 1993), which postulates that unregulated Aβ ion channels result in a loss of ionic homeostasis (primarily through a gain of $Ca^{2+}$) that eventually triggers neuronal dysfunction and cell death. *In vivo* evidence for this mechanism is, however, still rare, and a compound that would block pores and be active in mammalian animal models has not been reported yet. Thus, the original request by Arispe and coworkers that a useful strategy for drug discovery for treatment of AD should include screening compounds for their ability to block or otherwise modify Aβ channels is still left unsatisfied (Arispe *et al*, 1993).

In this work, we examine the efficacy of the diphenylpyrazole (DPP) compound anle138b in an animal model of Aβ deposition. Oral application of anle138b ameliorates Aβ-induced deficits in synaptic plasticity and memory formation. Using *in vivo* and *in vitro* approaches, we provide evidence that this effect is linked to the capacity of anle138b to reduce the conductivity of Aβ pores in lipid bilayer membranes. Although other mechanisms likely contribute to this effect, our data suggest the functional modulation of the membrane bound Aβ-oligomers as a mechanism for neuroprotection and support the idea that anle138b should be taken into clinical trials to treat aggregopathies, including AD.

## Results

### Synaptic plasticity and memory function in a mouse model for deposition of amyloid β peptides after oral treatment with anle138b

To initially test the potential of anle138b as therapeutic strategies to treat amyloid aggregation in Alzheimer's disease, we analyzed its effect in a *Drosophila* model for amyloid-induced neurotoxicity. We observed that treatment with anle138b improved survival times when compared to a vehicle-treated group (Appendix Fig S1). On the basis of these data, we decided to test the efficacy of anle138b in a mouse model for amyloid deposition. We like to state that none of the currently employed animal models for AD fully recapitulate the phenotypes seen in AD patients, and thus, care has to be taken when interpreting such data. In our study, we employed APPPS1Δ9 mice (Jankowsky *et al*, 2001), a well-established model for AD-linked amyloid deposition. Since in the patients therapeutic intervention is normally initiated only after the onset of amyloid plaque formation, we decided to test anle138b in two experimental cohorts. In the "pre-plaque group," treatment was initiated before the onset of pathology when mice were 2 months of age, while in the "post-plaque group" treatment was initiated after the onset of amyloid deposition and memory disturbances in 6-month-old mice (Fig EV1;

Jankowsky *et al*, 2004; Lalonde *et al*, 2005; Reiserer *et al*, 2007). In both cohorts, anle138b was continuously provided via food pellets. Thus, in the pre-plaque group, mice were subjected to anle138b or placebo treatment from 2 months of age, and electrophysiological, behavioral, and biochemical analyses were initiated at 6 months of age. A group of wild-type mice (WT) treated with anle138b served as an additional control. We first measured synaptic plasticity by analyzing hippocampal long-term potentiation (LTP). While robust hippocampal LTP at the Schaffer collateral synapse was observed in WT control mice treated with anle138b (Fig 1A), LTP was significantly impaired in APPPS1Δ9 mice that received placebo (Fig 1B). Notably, this LTP deficit was completely rescued in APPPS1Δ9 mice treated with anle138b (Fig 1C). These data suggest that oral application of anle138b protects against Aβ-induced impairment of hippocampal synaptic plasticity. To test whether the effect of anle138b on hippocampal plasticity also improved hippocampus-dependent memory function, another group of anle138b and placebo-treated mice were subjected to the Morris water maze test, a well-established paradigm to assay spatial memory in rodents (Morris, 1984). Anle138b-treated WT mice displayed robust spatial learning as indicated by decreasing escape latency throughout the 8 days of training (Fig 1D). In contrast, APPPS1Δ9 mice treated with placebo showed a significantly impaired escape latency (Fig 1D). This deficit was partially rescued in APPPS1Δ9 mice that received anle138b. Spatial reference memory was analyzed in a probe test performed after 8 days of training. While WT mice showed a significant preference for the target quadrant, no such effect was observed in placebo-treated APPPS1Δ9 mice (Fig 1E), confirming memory impairment in APPPS1Δ9 mice. In contrast, anle138b-treated APPPS1Δ9 mice displayed a significant preference for the target quadrant indicating restored spatial memory (Fig 1E). Swim speed was similar amongst the groups (Fig 1F). We also examined if anle138b would affect basal explorative behavior (Fig 1G) or basal anxiety (Fig 1H). No difference was found amongst the groups suggesting that oral administration of anle138b can protect APPPS1Δ9 mice from deteriorating hippocampal synaptic plasticity and hippocampus-dependent memory consolidation.

Encouraged by these findings, we investigated whether anle138b could also reinstate synaptic plasticity and memory function when significant amyloid deposition had already occurred employing the post-plaque group (Fig EV1). To this end, 6-month-old APPPS1Δ9 mice were treated with either anle138b or placebo for 4 months. Wild-type mice treated with anle138b served as an additional control group. Analysis was performed when mice were 10 months of age. In a first cohort, we measured hippocampal LTP. WT mice treated with anle138b showed robust LTP (Fig 2A), while LTP was significantly impaired in placebo-treated APPPS1Δ9 mice (Fig 2B). Notably, a complete restoration of hippocampal LTP was seen in APPPS1Δ9 mice treated with anle138b (Fig 2C). In conclusion, similar to the pre-plaque group treatment with anle138b had a marked ameliorating effect on LTP even after the onset of plaque deposition.

To analyze whether reinstatement of hippocampal plasticity would also correlate with improved memory function, we subjected mice to the Morris water maze test. WT mice treated with anle138b rapidly learned the task as indicated by reduced escape latency throughout the 8 days of training (Fig 2D). Placebo-treated 10-month-old APPPS1Δ9 mice display impaired spatial learning as indicated by the escape latency that did not significantly decrease

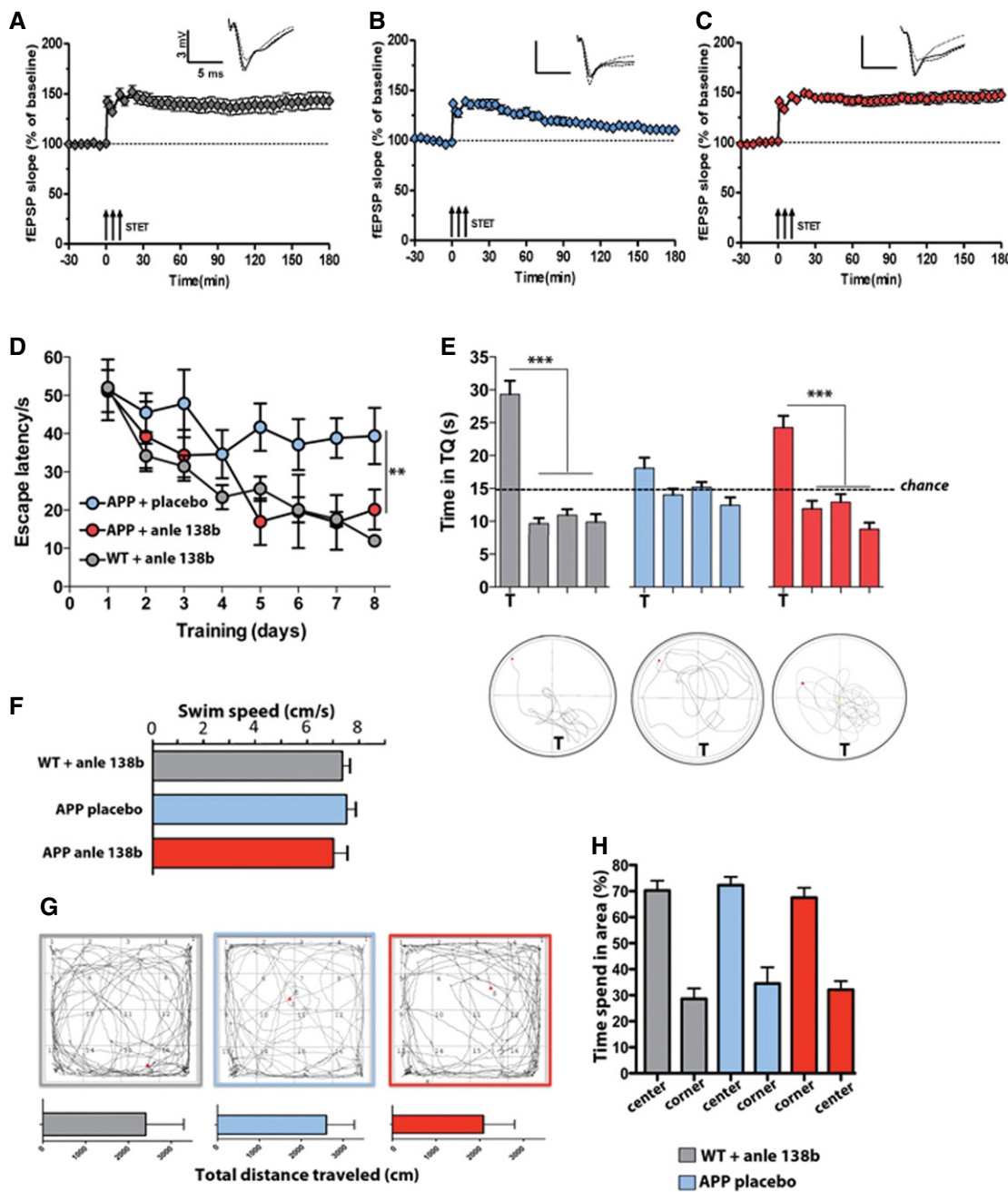

**Figure 1.  Anle138b rescues hippocampal LTP deficits spatial memory in the pre-plaque group.**

A    In wild-type mice of the pre-plaque group (treated from 2 to 6 months of age with anle138b), robust LTP that lasts for at least 3 h is elicited upon a strong tetanization (STET) (3 trains of 100 pulses at 100 Hz given 10 min apart, arrows) at the Schaffer collateral CA3-CA1 synapse (*t*-test, *P* = 0.00005; *n* = 16 comparing before vs. after STET).

B    LTP is not maintained in APPPS1Δ9 mice treated with placebo. Here, the potentiation declined to baseline after 3 h (*t*-test, *P* = 0.08; *n* = 20 comparing before vs. after STET).

C    APPPS1Δ9 treated with anle138b show a rescue of the LTP impairment (*t*-test, *P* = 0.0001; *n* = 23 comparing before vs. after STET).

D    Escape latency in the Morris water maze test is impaired in placebo- but not in anle138b-treated APP mice (one-way ANOVA *F* = 16.01, **P* = 0.0008; *n* = 15/group).

E    Probe test performed 24 h after the last training session. The lower panel shows representative swimming path during the probe test. T = target quadrant compared vs. other quadrants (*t*-test, ***P* = 0.00002; *n* = 15/group).

F    Average swim speed during water maze training (*n* = 15/group).

G    Explorative behavior in the open field test. Upper panel: Representative motion tracks during the test session. Lower panels show the total distance travelled during the 5-min test session (*n* = 15/group).

H    Bar graph showing the time spent in the center vs. the corner of the open field (*n* = 15/group).

Data information: Error bars indicate SEM.

during the training (Fig 2D). When compared to the placebo group, APPPS1Δ9 mice treated with anle138b showed improved spatial learning (Fig 2D). A probe test was performed after 8 days of training. WT mice treated with anle138b showed a significant preference for the target quadrant indicative of intact spatial reference memory (Fig 2E). Placebo-treated APPPS1Δ9 mice exhibited severely impaired memory function (Fig 2E) and displayed no target preference (Fig 2E). In APPPS1Δ9 mice treated with anle138b (Fig 2E), target preference was improved significantly but did not reach WT levels. Of note, swim speed was not different between the experimental groups (Fig 2F). Explorative behavior (Fig 2G) and basal anxiety (Fig 2H) were measured in the open field test. There was no significant difference amongst groups. We also tested whether anle138b treatment would affect memory function in wild-type mice but did not observe any significant difference (Appendix Fig S2). Thus, oral administration of anle138b partially restores hippocampal plasticity and memory function in APPPS1Δ9 mice even at an advanced stage of pathology.

### Anle138b reinstate transcriptional homeostasis and ameliorates amyloid pathology

Pathological alterations often lead to aberrant changes in transcriptional plasticity indicating deregulated cellular homeostasis (Fischer, 2014a). In support of this, numerous studies demonstrated that AD pathogenesis correlates with altered gene expression in various brain regions (Fischer, 2014b; Benito *et al*, 2015; Matarin *et al*, 2015). Moreover, monitoring gene expression changes can inform about the efficacy of therapeutic intervention (Benito *et al*, 2015). In keeping with this idea, we performed RNA sequencing from hippocampal tissue dissected from mice in the pre- and post-plaque (Fig EV1) cohorts. Notably, there were no differences in hippocampal gene expression when comparing WT placebo- vs. WT anle138b-treated mice, suggesting that anle138b has no direct effect on transcriptome plasticity. Thus, we first compared gene expression in placebo-treated WT and placebo-treated APPPS1ΔE9 mice of the pre-plaque group. We identified 202 differentially expressed genes of which 73 were up- and 129 were down-regulated (Fig 3A; Dataset EV1). Pathway analysis shows that down-regulated genes are linked to reduced energy metabolism, mitochondria function, cytoskeleton integrity, and synaptic plasticity, while pathways linked to cell growth were increased (Fig 3B). These data are in line with previous reports of gene expression changes in AD (Benito *et al*, 2015; Matarin *et al*, 2015) and were confirmed via qPCR (Appendix Fig S3A).

Because anle138b restored LTP and spatial memory functions in pre-plaque mice (see Fig 1), we hypothesized that the transcriptome of APPPS1ΔE9 mice treated with anle138b may more closely resemble that of wild-type mice. Indeed, a wild-type-like gene expression profile is to a large extent reinstated in anle138b-treated APPPS1ΔE9 and only 27 genes were deregulated when comparing wild-type mice treated with anle138b vs. APPPS1ΔE9 mice treated with anle138b groups (Fig 3A). Thus, none of the pathways deregulated in response to amyloid pathology (Fig 3B) remained significant after anle138b treatment. Accordingly, no enrichment for any specific pathway could be detected. The possibility remained that anle138b treatment may simply reduce the expression levels of the APP and PS1 transgenes specifically in APPPS1ΔE9. However, the RNA-seq data showed that APP and PS1 are increased in placebo-treated

APPPS1ΔE9 mice when compared to placebo-treated wild-type mice (see Dataset EV1). Similarly, APP and PS1 transgenes were elevated in anle138b-treated APPPS1ΔE9 mice compared to anle138b-treated wild-type mice (see Dataset EV1). These data were confirmed via qPCR showing that the expression of APP and PS1 transgenes is similar in placebo- and anle138b-treated APPPS1ΔE9 mice (Fig 3C); the gene expression data are in line with the electrophysiological and behavioral findings. It also shows that in the pre-plaque group, anle138b treatment reinstates cellular homeostasis in the hippocampus of APPPS1ΔE9 mice.

We next analyzed hippocampal gene expression in the post-plaque group (Fig EV1). When comparing WT placebo and APPPS1ΔE9 placebo mice, we found 130 differentially expressed genes of which the majority (124) were up-regulated (Fig 3D), a finding confirmed via qPCR (Appendix Fig S3A). The comparison of anle138b-treated WT and anle138b-treated APPPS1ΔE9 mice revealed 220 differentially expressed genes, 207 up-regulated, and 13 down-regulated (Fig 3D). Around half of these, 103 were also deregulated in the comparison WT placebo vs. APPPS1ΔE9 placebo (Fig 3E), a finding confirmed via qPCR (Appendix Fig S3B). Pathway analysis revealed that gene expression changes almost exclusively represent an induction of neuroinflammatory processes (Fig 3F) in response to the APP and PS1 transgenes and this induction is not changed by treatment with anle138b. These data suggest that anle138b treatment does not have a major impact on inflammatory processes when given at a stage of advanced amyloid pathology. Since anle138b treatment nevertheless reinstated hippocampal synaptic plasticity and also partially restored memory function, these findings indicate that the therapeutic efficacy of anle138b is most likely not solely due to the dampening of amyloid-induced inflammation.

A previous study reported that anle138b exhibits therapeutic effect in TauP301S mice, a mouse model for Tau pathology (Wagner *et al*, 2015). Taking into account that Tau and amyloid pathology represent the two major causative factors for AD, we wondered if anle138b would affect brain homeostasis as measured by hippocampal gene expression also in TauP301S mice. To this end, we obtained hippocampal brain tissue from the same experiment using anle138b or vehicle-treated wild-type and TauP301S mice as described by Wagner *et al* (2015). The experimental design employed by Wagner *et al* is similar to our pre-plaque group, since anle138b was feed to mice upon weaning. Similar to our data, feeding anle138b to wild-type mice had a neglectable effect on hippocampal gene expression when compared to vehicle fed wild-type mice (Appendix Fig S4, Dataset EV1). The comparison of vehicle-treated wild-type to vehicle-treated TauP301S mice revealed 257 differentially expressed genes that mainly represent increased pathways linked to neuroinflammation (Appendix Fig S4). Of note, this gene expression signature was significantly ameliorated in anle138b-treated TauP301S mice and only 16 genes were differentially expressed when comparing vehicle- vs. anle138b-treated TauP301S mice (Appendix Fig S4). These data allow for a number of interesting conclusions. First, at the hippocampal gene expression level, the overexpression of the human Tau gene carrying the P301S mutation leads to inflammatory response but in contrast to the data observed in APPPS1ΔE9 mice has very limited impact on the expression of genes linked to synaptic plasticity and learning and memory. Second, treatment with anle138b in TauP301S mice ameliorates the gene expression

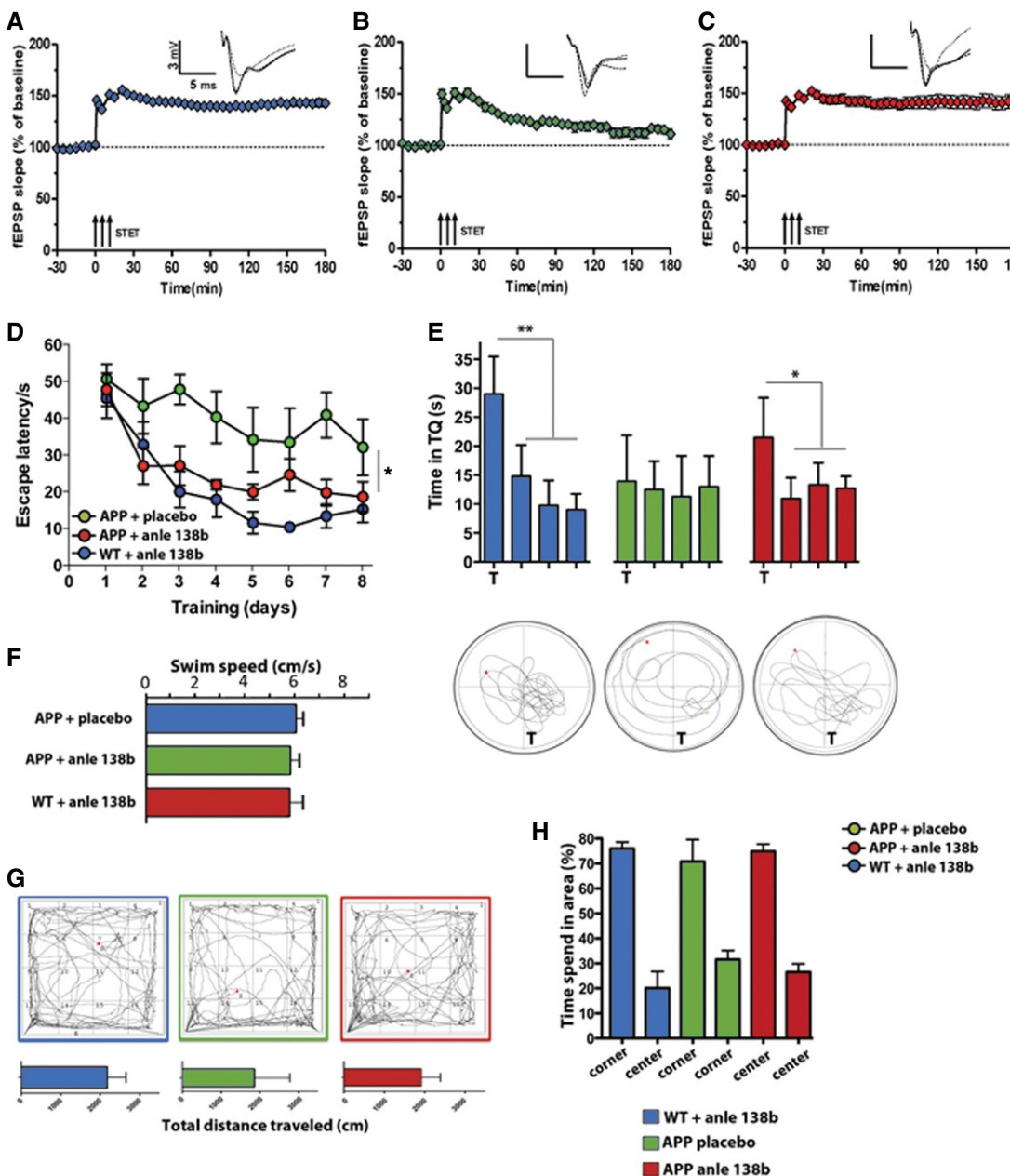

**Figure 2. Anle138b rescues hippocampal LTP deficits and spatial memory in the post-plaque group.**

A   Wild-type mice of the post-plaque group (treated from 6 to 10 months of age) display robust LTP upon STET (arrows) that was maintained throughout the recording session (*t*-test, *P* = 0.00006; *n* = 30 comparing before vs. after STET).

B   Lasting LTP induced by STET was not observed in 10-month-old APPPS1Δ9 placebo-treated mice. The potentiation decayed to baseline after 3 h (*t*-test, *P* = 0.16; *n* = 19 comparing before vs. after STET).

C   Treatment with anle138b starting at 6 months of age rescues LTP deficit in 10-month-old APPPS1Δ9 mice (*t*-test, *P* = 0.0003; *n* = 20 comparing before vs. after STET).

D   Escape latency in the Morris water maze test is impaired in placebo-treated APP mice and partially restored to WT levels in anle138b-treated APP mice (one-way ANOVA, *F* = 35.94, *P* = 0.00006; *P* = 0.0309 for APP + anle138b vs. APP + placebo and *P* = 0.4 for APP + placebo vs. WT + anle138b; *n* = 7/group).

E   Probe test performed 24 h after the last training session. The lower panel shows representative swimming path during the probe test. T = target quadrant compared vs. other quadrants (*t*-test, **P* = 0.0007, *P* = 0.0031; *n* = 7/group).

F   Average swim speed during water maze training (*n* = 7/group).

G   Explorative behavior in the open field test. Upper panel: Representative motion tracks during the test session. Lower panels show the total distance travelled during the 5-min test session (*n* = 7/group).

H   Bar graph showing the time spent in the center vs. the corner of the open field (*n* = 7/group).

Data information: Error bars indicate SEM. In panels (A–C), triplets of arrows represent STET applied for inducing L-LTP. Insets in each graph represent typical fEPSP traces recorded 15 min before (dotted line), 30 min after (broken line), and 3 h after (full line) STET.

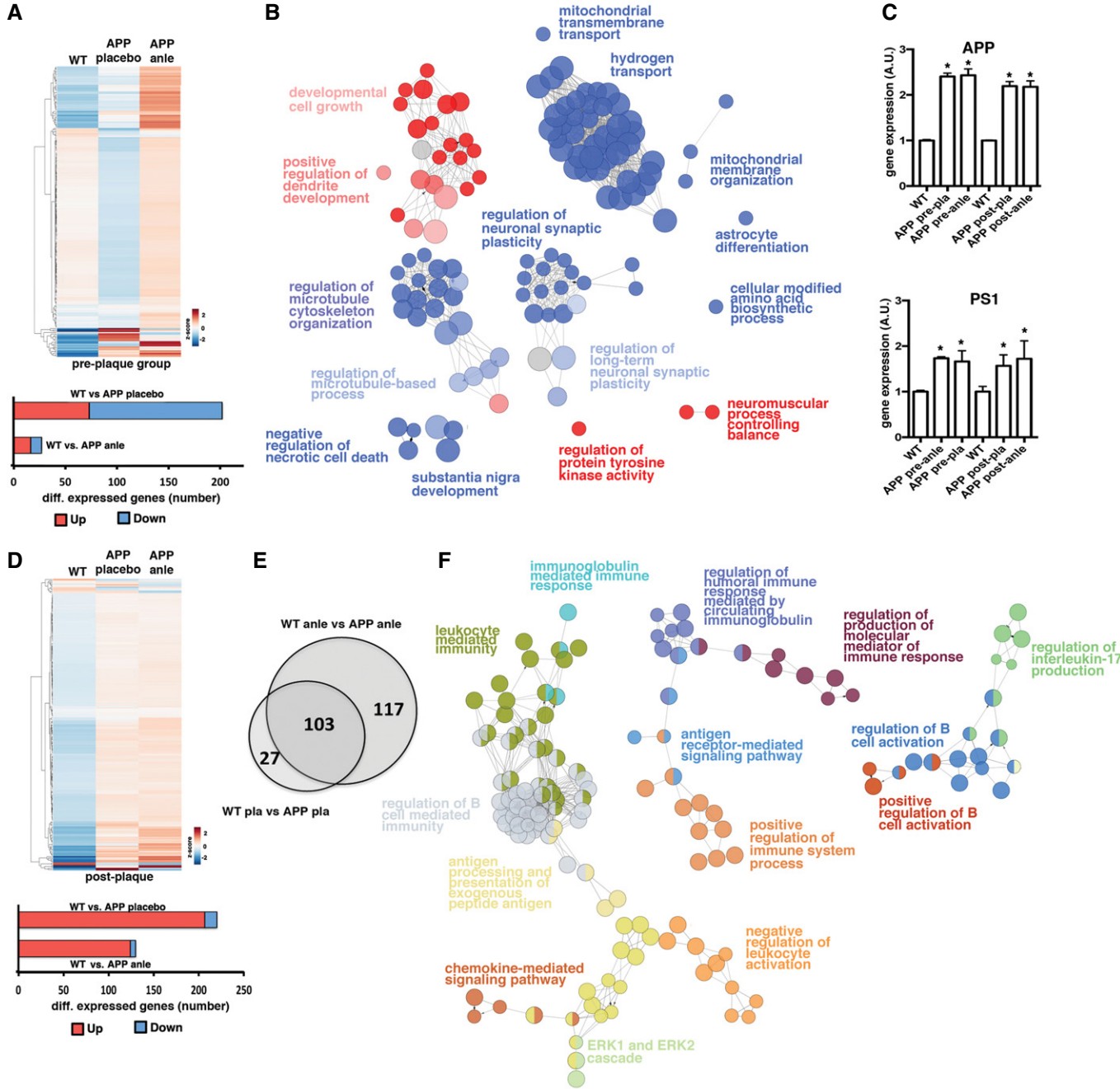

**Figure 3.  Hippocampal transcriptome analysis in anle138b-treated mice.**

A   Upper panel: Heat map showing differentially expressed genes in placebo-treated WT ($n$ = 4), placebo-treated APPPS1ΔE9 mice ($n$ = 4), and anle138b-treated APPPS1ΔE9 mice ($n$ = 4) of the pre-plaque group. Note that a WT-like gene expression profile is to a large extent reinstated in APPPS1ΔE9 mice upon treatment with anle138b. Lower panel: Bar graph showing the number of up and down-regulated genes.

B   Pathways down-regulated (blue) or up-regulated (red) in 6-month-old APPPS1ΔE9 mice when compared to age-matched WT controls (pre-plaque group; see Fig EV1).

C   Expression of APP and PS1 genes in WT and APPPS1ΔE9 transgenic mice, placebo- or anle138b-treated in pre- and post-plaque groups. Note that anle138b treatment does not affect the expression level of APP or PS1. *Post hoc* analysis revealed a significantly increased expression of APP and PS1 in transgenic mice when compared to wild-type mice (*$P$ < 0.05). Error bars indicate SEM.

D   Upper panel: Heat map showing differentially expressed genes in placebo-treated WT ($n$ = 4), placebo-treated APPPS1ΔE9 mice ($n$ = 5), and anle138b-treated APPPS1ΔE9 mice ($n$ = 5) of the post-plaque group. Lower panel: Bar graph showing the number of up- and down-regulated genes.

E   Venn diagram showing that 10-month-old mice treated with placebo or anle138b show very similar changes in hippocampal gene expression (post-plaque group; see Fig EV1).

F   Pathway analysis based on the 103 genes commonly increased in placebo- and anle138b-treated APPPS1ΔE9 mice at 10 months of age. Note that the pathways are exclusively linked to neuroinflammation.

changes, at least when treatment is initiated before the onset of pathology, further confirming that treatment with anle138b helps to reinstate hippocampal homeostasis.

Given that anle138b was shown to counter aggregation in models for α-synuclein and prion toxicity (Wagner *et al*, 2013), it is likely that at least part of the therapeutic effect observed in this study is due to anle138b interfering with amyloid aggregation. To test this hypothesis, we prepared brain slices for histochemical analysis. We used thioflavin S, a common stain to quantify Aβ plaques (Fig 4A). We first assayed the amyloid plaques pathology in the hippocampus and cortex in the pre-plaques group. Since no amyloid pathology was detectable in WT mice, these mice served as negative control. We observed a significant reduction in the number of plaques and the total area covered by plaques in anle138b-treated mice (Fig 4A and B). Similar results were obtained when we analyzed plaque load in the post-plaques group (Fig 4B and C), although the pathology was generally more severe in the post-plaque group. Thus, oral administration of anle138b reduces amyloid pathology when given before or after the onset of pathology.

### Biophysical characterization of Aβ in the presence of anle138b

To better understand the mechanisms that underlie the therapeutic effect of anle138b, we analyzed its impact on Aβ-induced pore formation (Arispe *et al*, 1993). To this end, we employed black lipid membranes (BLM) using a mixture of POPE and DOPS in a 1:1 ratio or oxidized cholesterol/n-decane, an assay in which the current passing through a membrane is a measure of membrane integrity (Appendix Fig S5A). We first established that the conductance and the morphology of POPE and DOPS lipids, as measured by atomic force microscopy (AFM), lipid bilayers are not affected in the presence of anle138b or Aβ peptides (Appendix Fig S6).

When we examined the effect of Aβ$_{1-42}$ in DOPS/POPE (1:1) membranes, we observed a stepwise growth of bulk membrane conductance. Such activity is indicative of the combined action of many individual pores (Fig 5A). Pore "stacking" is likely the result of prolonged open lifetimes and the formation and/or opening of additional conducting pores. Anle138b-doped membranes demonstrated fewer simultaneously active pores and significantly reduced bulk conductance (Fig 5A and B, Appendix Fig S5) compared to membranes lacking the anle138b compound (Fig 5A and B, Appendix Fig S5). Our results indicate that treatment with anle138b alters the pore dynamics, resulting in less stable and shorter lived "open" pores. Decreased pore stability leads to a reduction in the total number of simultaneously conducting pores and significantly decreased conductance across the membrane. AFM data revealed that anle138b treatment did not affect the surface structure of Aβ$_{1-42}$ pores (Fig EV2), suggesting that anle138b does not simply prevent Aβ$_{1-42}$ from entering lipid bilayer membranes and forming pores. Rather anle138b appears to render conducting Aβ pores to non-conducting ones—likely through structural change to the membrane embedded region of Aβ$_{1-42}$—thereby providing one possible mechanism by which anle138b ameliorates LTP and learning deficits in APPPS1ΔE9 mice. Similar effects were observed when the conductance measurements were repeated in oxidized cholesterol (Appendix Fig S5B).

To provide *in vivo* evidence for pore formation, we treated primary hippocampal neurons with Aβ$_{1-40}$ monomers or oligomers

in the absence or presence of anle138b and assayed membrane integrity. While the addition of Aβ$_{1-40}$ oligomers but not Aβ$_{1-40}$ monomers significantly damaged membrane integrity, this effect was reversed by anle138b (Fig 5C). This restoration is not simply due to an anle138b-mediated reduction in cell death, since the cell viability of hippocampal neurons measured in the MTT assay was identically affected by Aβ$_{1-40}$ monomer and oligomer treatment in our experimental setting (Fig 5D). In this context, it is important to state that different protocols for Aβ treatment of neurons have been used to study various effects of Aβ on neuronal plasticity and integrity. The concentration of Aβ oligomers employed in our study is within the range of these studies. Nevertheless, such data always need to be interpreted with care since a cell culture system cannot not fully recapitulate the situation observed in human patients. Interestingly, anle138b treatment was also able to ameliorate the effect of toxic Aβ species on membrane integrity if added after neurons had been incubated with Aβ oligomers and already exhibited impaired viability, which is in line with our data showing that anle138b completely ameliorated LTP and partially restored memory defects in the post-plaque group (Appendix Fig S7A and B). We also employed another experimental system to test the effect of Aβ$_{1-42}$ oligomers and anle138b on biological membranes. To this end, we employed the cytochrome release assay (CRA) on isolated mitochondria that were exposed to either α-synuclein, tau, or Aβ$_{1-42}$ oligomer in the presence or absence on anle138b. Our data reveal a substantial damage to mitochondrial membranes in all conditions that is attenuated by anle138b (Appendix Fig S7C), suggesting that anle138b has general effect on membrane integrity.

## Discussion

In this work, we investigated the effect of anle138b in the established APPPS1ΔE9 mouse model of AD. The APPPS1ΔE9 mouse model is characterized by dysfunction of neurons and detected by memory decline after 4 months and severe plaques formation after 6 months (Kummer *et al*, 2014). The most important findings of the present study were that anle138b treatment completely restored hippocampal LTP in the pre- and post-plaque groups of APPPS1ΔE9 mice. In line with this observation, spatial reference memory was fully or partially restored in the pre- and post-plaque groups, respectively. There have been numerous pre-clinical studies aiming to restore synaptic plasticity and memory function in mouse models for amyloid pathology. These range from therapeutic approaches that aim to modulate causative factors including Aβ-toxicity via antibody-based therapies (Selkoe & Hardy, 2016), small molecules that modulated APP processing (Vassar *et al*, 2014), small molecules that are thought to target toxic amyloid species directly (Doig & Derreumaux, 2015) to symptomatic treatments not directly targeted toward amyloids (Fischer, 2016). Anle138b belongs to the first category. Its therapeutic effect resembles that of other small molecule drugs. For example, epigallocatechin gallate (EGCG) was shown to affect Aβ toxicity by redirecting toxic Aβ-structures into off-pathway oligomers (Ehrnhoefer *et al*, 2008; Bieschke *et al*, 2010). This agent was able to ameliorate spatial memory deficits in APP mice after the onset of amyloid deposition (Liu *et al*, 2014). In this context, it is interesting to note that when administered via dry food anle138b

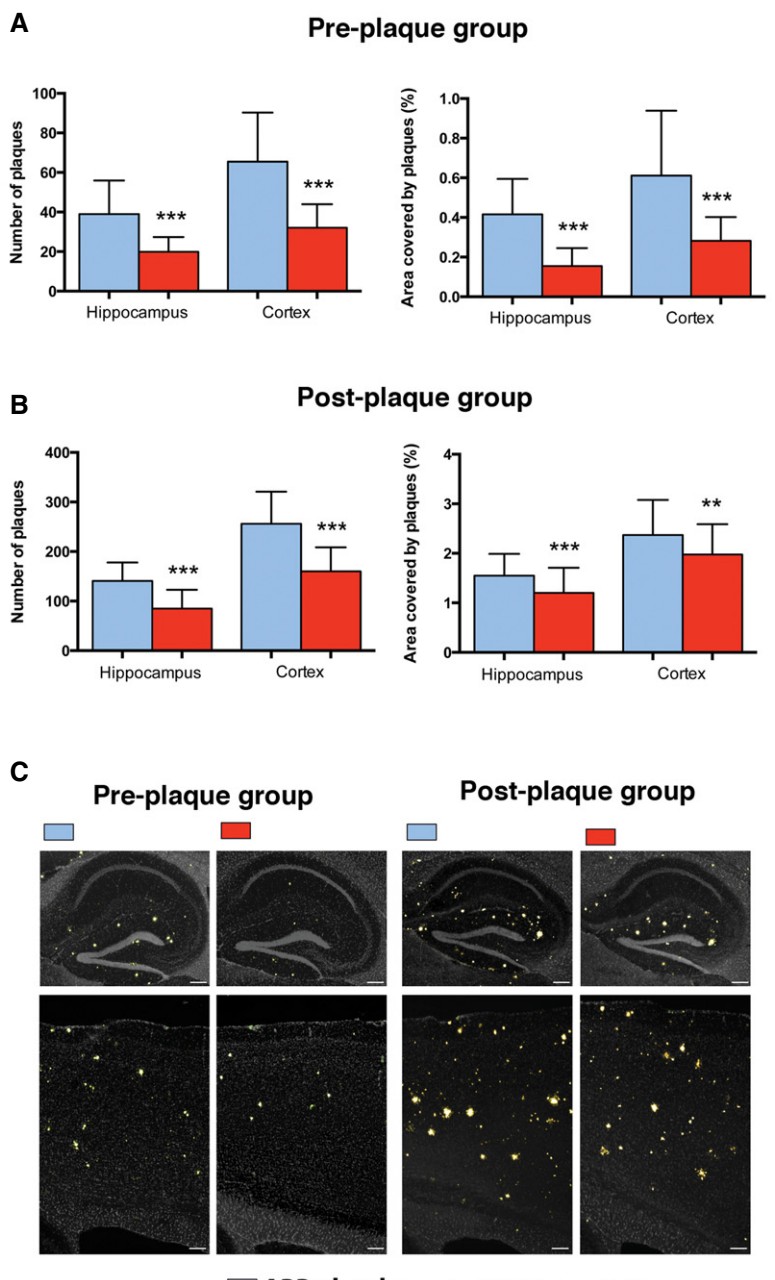

**Figure 4.  Amyloid plaque pathology is ameliorated by anle138b treatment in the pre-plaque (A) and post-plaque (B) groups.**

A   Reduced amyloid pathology as indicated by reduced number of plaques (left panel) and reduced area covered by plaques (right panel) in the pre-plaque group (*t*-test ***$P$ = 0.00003; *n* = 5/group).

B   Reduced amyloid pathology as indicated by reduced number of plaques (left panel) and reduced area covered by plaques in the post-plaque group (*t*-test ***$P$ = 0.00007, **$P$ = 0.005; *n* = 5/group).

C   Representative images showing thioflavin S staining in the hippocampus and cortex of mice of the pre- and post-plaque groups. Scale bar = 200 μm.

Data information: Error bars indicate SEM.

was previously found to be taken up to reach levels of 100 μM in the brain (Wagner *et al*, 2015). Anle138b metabolites are very hydrophilic, and they were detected in multiple organs but were absent in brain (Wagner *et al*, 2013). This indicates that anle138b is not metabolized in the brain. Anle138b was also found to be non-toxic in mice up to a dose of 2 g/kg. Mice receiving similar concentrations of anle138b as used in our study lived without any detectable toxicity, even when the drug was given longer than a year (Wagner *et al*, 2015). Also in our study, no negative or positive effects of anle138b on WT mice were observed.

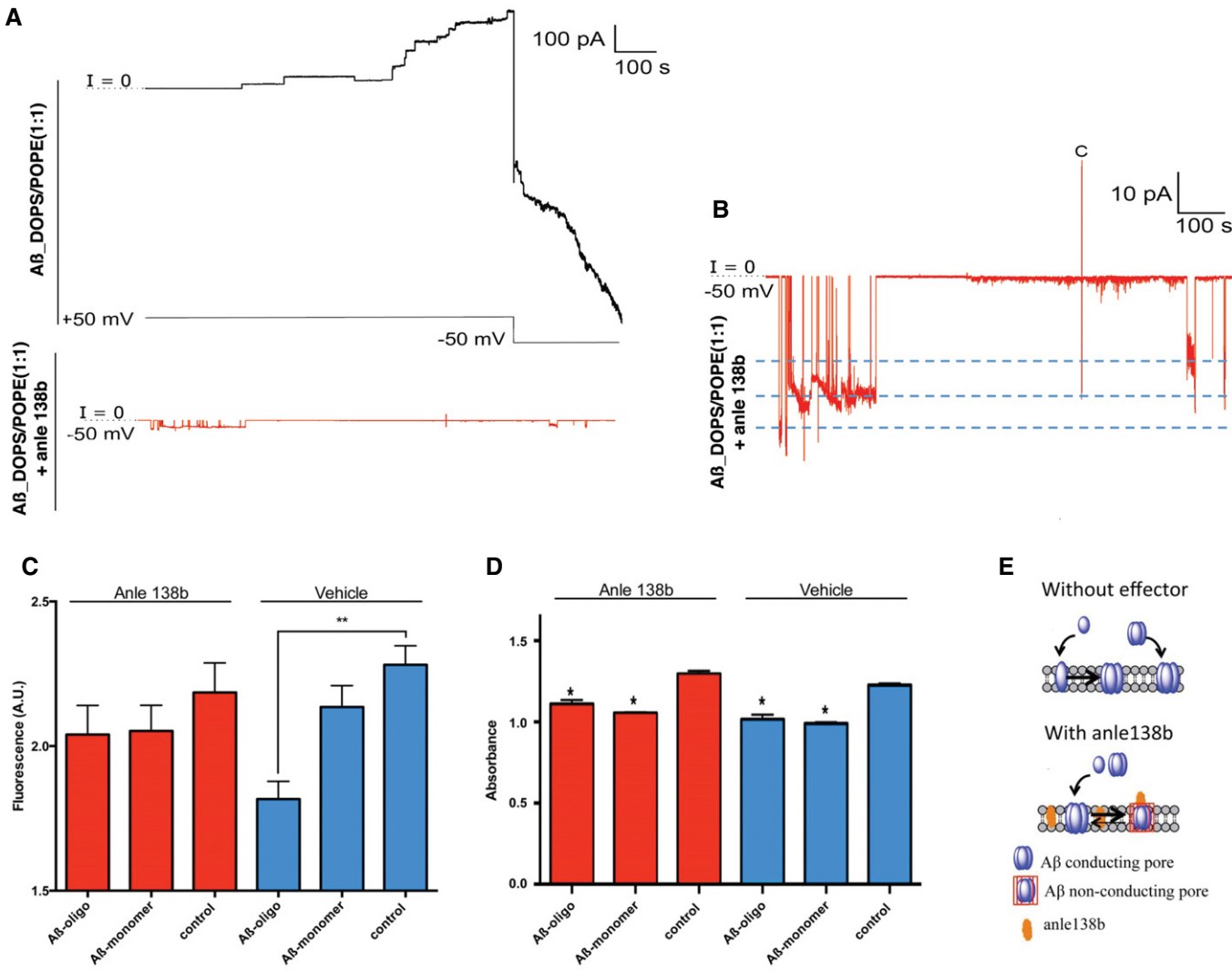

**Figure 5. Anle138b ameliorates Aβ$_{1-40}$ and Aβ$_{1-42}$-induced membrane integrity.**

A  Aβ$_{1-42}$ induces pore-like step ionic current increases across lipid bilayer membranes and grows with inverted sign after voltage inversion to amplifier saturation current. In the presence of anle138b, the current remains at low amplitude and does not increase beyond 30 pA.

B  Expanded trace of anle138b-treated membrane presented in (A). Discreet conductance levels are highlighted suggesting multiple opening and closing events for three individual pores.

C  Hippocampal neurons (DIV 10) were treated with anle138b (1 μM) or vehicle before Aβ$_{1-40}$ oligomers or monomers were added (10 μM, *n* = 4/group). After 48 h, membrane integrity was measured as fluorescence intensity using a CyQUANT assay (Thermo Fisher). In the vehicle group, membrane integrity was significantly impaired when treated with Aβ oligomers comparing with control neurons or Aβ monomer-treated neurons, anle138b-treated neurons did not exhibit a difference between addition of Aβ monomers or Aβ oligomers (*t*-test, **P = 0.007).

D  Same experimental setting as in (C) but cell viability was measured using the MTT assay (*t*-test, *P = 0.002 vs. control). No difference in cell viability was observed for Aβ monomer or oligomer treatment in the absence or presence of anle138b.

E  Schematic of potential mechanisms of activity inhibition for anle138b. In the absence of anle138b, Aβ monomers and/or oligomers insert in the membrane and form conducting pores. Treatment with anle138b renders these conductive channels inactive most probably due to reduced life time of the open state.

Data information: Error bars indicate SEM.

In addition to the restoration of hippocampal LTP and spatial reference memory, we also observed restoration of physiological, wild-type-like gene expression profile in the pre-plaque group, indicating that hippocampal cells reinstate homeostasis. In APPPS1ΔE9 mice that received normal chow, genes linked to metabolic function and neuronal plasticity were markedly down-regulated. Feeding with anle138b-containing chow completely reversed this effect on gene expression. Even in the placebo-treated pre-plaque group, we observed little evidence for neuroinflammatory processes, which is in contrast to the data from the post-plaque group. In fact, the gene expression changes observed in the placebo-treated post-plaque group were dominated by increased inflammation. These data suggest that in APPPS1ΔE9 mice, the decline of synaptic plasticity precedes inflammatory processes. Of note, anle138b treatment had

no effect on pathological gene expression pattern in the post-plaque group, yet hippocampal LTP was completely restored and spatial memory was partially restored. These data suggest that therapeutic strategies that aim to reduce amyloid toxicity—at least in the APPPS1ΔE9 model—may have little influence on neuroinflammation when applied at an advanced stage of the disease. Nevertheless, a significant therapeutic effect could be observed in the post-plaque group. This might be explained by removal of toxic Aβ species which can lead to the restoration of synaptic function in neurons even in a detrimental inflammatory environment. These data are highly interesting, since one argument for the failure of clinical trials targeting amyloid deposition or modulation of Aβ cleavage is that such treatments are ineffective when given at an advanced stage of the diseases. Our data suggest, however, that anle138b targets Aβ-related pathological events that allow recovery of synaptic function even if secondary pathological events such as inflammation persist.

The finding that anle138b treatment ameliorates synaptic plasticity and learning deficits in APPPS1ΔE9 is of utmost importance. These data are in line with previous observations showing that the same compounds have beneficial effects in animal models for Creutzfeld-Jakob, Parkinson's disease, and Tau pathology. Specifically, the latter finding is intriguing, since Tau and Aβ pathology are believed to be the main causative factors of AD pathogenesis. While Wagner et al (2015) showed that anle138b in Tau P301S mice rescues synaptic and neuronal loss, we could further substantiate these data by showing that anle138b treatment also ameliorates defects in hippocampal transcriptome plasticity in the same Tau P301S mice. Taken together, these data suggest the revealing possibility that anle138b is able to ameliorate Tau and Aβ pathology. Therefore, to the best of our knowledge, anle138b would be the first small molecule that has a direct effect on the two major hallmarks of AD.

These data are also important from a translational point of view. Taking into account that the various animal models for AD only recapitulate part of the pathogenesis observed in human patients, it is intriguing that anle138b was able to rescue disease progression in various different AD models and models for aggregopathies, raising the hope that anle138b could also be effective in human patients.

The mechanism by which toxic Aβ species induce synaptic dysfunction and neurotoxicity is still unresolved. Proposed mechanisms include NMDA receptor endocytosis (Shankar et al, 2007), α-amino-3-hydroxy-5-methyl-4-isoxazole propionic acid surface receptor modulation (Querfurth & LaFerla, 2010), and Aβ pore formation (Arispe et al, 1993). Moreover, the presence of amyloid plaques was shown to change structural plasticity of neurons (Spires & Hyman, 2004). Nevertheless, amyloid plaques are unlikely the sole reason for disturbed synaptic function and memory consolidation in AD and it has even been suggested that they present a compensatory mechanism of the brain in order to deal with toxic Aβ species (Selkoe & Hardy, 2016). This is further supported by our findings that anle138b treatment in the post-plaque group was able to restore hippocampal LTP and spatial reference memory, although the plaque load was even higher when compared to placebo-treated APPPS1ΔE9 mice of the pre-plaque group suggesting that the reduction in amyloid plaque load cannot be the main mechanisms by which anle138b exerts its therapeutic action. The pore formation hypothesis has long been proposed but tested experimentally to a

very limited degree, which is also due to the fact that an array of various in vitro and in vivo methods needs to be combined to address this issue. We focused on the question whether anle138b modulates the pore forming activity of Aβ because synaptic function relies on the integrity of membranes and their ability to modulate ion fluxes in a voltage-dependent way. This pore-based mechanism has been envisaged to underlie impaired neuronal function and cell death (Arispe et al, 1993). Modulation of pore formation was also implicated in cell-based models for EGCG mode of action (Diaz et al, 2009) and of its congeners MRS2481 and MRS2485 (Diaz et al, 2009). The observed stepwise growth of bulk membrane conductance in the presence of Aβ (without anle138b) indicates that activity increases through the combined action of many individual pores.

Numerous molecular dynamics (MD) simulations (Jang et al, 2008; Arce et al, 2011; Gillman et al, 2014) and NMR spectroscopy-based investigations (Sarkar et al, 2014) of Aβ pores have pointed to a β-barrel structure for the intramembrane region of the pore. It was previously shown that Aβ containing a point substituted proline (F19P) showed pore structure by AFM but did not demonstrate ionic conductance (Capone et al, 2012; Connelly et al, 2012b). The chemical structure of proline introduces a "kink" in the peptide's secondary structure, which is known to disrupt β-sheet formation. MD simulations of F19P barrel structure showed that β-sheet destabilization led the highly charged N-terminal regions to bind at the peptide mouth and collapsed the pore (Umehara et al, 2010; Connelly et al, 2012a). Our data are consistent with such a model and suggest that anle138b induces a conformational change within Aβ pores that greatly reduces or, in most pores eliminates, ionic flux. Steric blockage by anle138b without a conformational change of the pore is energetically unfavorable due to the hydrophobicity of anle138b, as this scenario would require anle138b to be in contact with water molecules inside the pore. The above-mentioned mechanisms provide further insight into the method of action in preventing pore activity and reducing Alzheimer's pathogenicity (Fig 5E).

It has to be mentioned, however, that all of the above-described experiments are based on in vitro approaches using lipid bilayers. In vivo evidence further supporting the notion that anle138b counteracts the detrimental effect of toxic Aβ species on membrane integrity stems from our finding that Aβ$_{1-40}$ administration to cultured hippocampal neurons compromised membrane integrity, an effect which was attenuated by anle138b. Our interpretation that conversion of conducting to non-conduction Aβ pores is one possible mode of anle138b action. It may also explain that anle138b ameliorated all tested disease phenotypes in the pre-plaque group, while in the post-plaque group only LTP was completely restored. Hence, in the post-plaque group, the presence of toxic Aβ species may already have induced secondary processes such as inflammation that persistently affects memory function even if membrane integrity and LTP are restored in neurons. Yet, we like to reiterate that while our data point to the existence of Aβ pores in vivo, a definite proof is still missing. Further evidence in support of the Aβ pores stems from a study in which Aβ oligomers induce single-channel calcium fluorescence transients in Xenopus oocytes (Demuro et al, 2011). Nevertheless, we cannot exclude that anle138b affects other cellular processes than Aβ-mediated conducting pores in membranes. Thus, it is also possible for example that in vivo Aβ sequesters membrane lipids, thereby affecting membrane integrity, which is then prevented by the action of anle138b.

    

In conclusion, our data show that anle138b can reinstate synaptic plasticity and memory function in a mouse model for amyloid pathology via mechanisms that—at least in part—appear to involve the blockage of Aβ-induced pores in membranes. Careful analysis of this activity indicates that the oligomers are still in the membrane, but pores have a changed conductivity profile, mainly staying open for shorter time and lacking the possibility of building up large currents as seen when anle138b was absent. Taking into account that anle138b was also shown to ameliorate disease phenotypes in a mouse model for Tau pathology, thus being to the best of our knowledge one of the first compounds that seems to causatively interfere with both of the two major hallmarks of AD, we suggest that anle138b to further be validated in clinical trials for its potential to treat AD and perhaps other aggregopathies.

# Materials and Methods

## Mouse experiments

The APP(Swe)/PS1EΔ9 (henceforth called: APPPS1Δ9) mouse model of AD was used for this study. The age of the animals is indicated for each experiment. In brief, mice of the pre-plaque group were analyzed at 6 months of age, while mice of the post-plaque group were 10 months of age. Upon completion of treatment, these and control mice underwent cognitive assessment by a behavior battery of tests. Electrophysiology, RNA sequencing, and histochemical analysis of plaque burden were also assessed. All mice were male and were maintained on a C57BL/6 background. They were kept in a 12-h dark/light cycle and housed in groups under constant standard conditions of temperature and humidity. Mice had *ad libitum* access to food and water. Animal care and handling were carried out in compliance with the Declaration of Helsinki and approved by local ethical committees.

## Anle138b treatment

In order to investigate the prophylactic effect of anle138b, we treated healthy, plaque-free, adult, APPPS1Δ9 mice with placebo- or anle138b-containing dry food pellets for 4 months from 2 to 6 months of age (pre-plaque group; Fig EV1). Age- and sex-matched wild-type littermates were also treated and served as controls. Similarly, in order to investigate the therapeutic effect of anle138b, we treated symptomatic APPPS1ΔE9 mice and treated them for 4 months from 6 to 10 months of age (post-plaque group; Fig EV1). Controls were age- and sex-matched wild-type littermates treated with anle138b or placebo. Anle138b was administered orally. Dry food pellets were prepared containing 2g anle138b per kg food (SSNIFF). This amounted to an estimated daily dose of 500 mg/kg (at an approx. 6-g daily food consumption and a 25-g average body weight). Based on pharmacokinetic studies, 40–70 μM anle138b reached the brain during most of the wake phase (Wagner *et al*, 2015). Placebo food was prepared from the same batch but without anle138b (SSNIFF). Of note, our previous PK studies in mice have shown that after a single bolus the half-life of anle138b in the brain is approximately 4 h (Wagner *et al*, 2013).

## Morris water maze (MWM)

Mice were single-caged and brought into the testing room at least 1 week prior to the beginning of the experiment to allow them to acclimate. In the MWM, mice were trained to find a submerged 10 × 10 cm platform in a pool (1.10 m in diameter) of milky water using spatial cues by the pool sides as orientation points. Mice were trained over 8 consecutive days with four trials per day per mouse. Time and path to platform were tracked and recorded (TSE systems). On day 9 (probe test), the platform was removed and quadrant preference was recorded (target quadrant being the one where the platform was previously located).

## Open field

Mice were allowed to spend 5 min in an open arena (40 × 40 cm). Path length while exploring and time in the center or corners were quantified using the VideoMot2 System (TSE).

## RNA sequencing

Library preparation and cluster generation for mRNA sequencing were performed according to Illumina standard protocols (TruSeq, Illumina). Libraries were quality-controlled and quantified using a Nanodrop 2000 (Thermo Scientific), Agilent 2100 Bioanalyzer (Agilent Technologies), and Qubit (Life Technologies). Data will be made available upon publication via GEO (accession number GSE104424). Base calling from raw images and file conversion to fastq-files was achieved by Illumina pipeline scripts. Subsequent steps included quality control (FastQC, https://www.bioinformatics. babraham.ac.uk/projects/fastqc/), mapping to reference genome (mm10, STAR aligner v2.3.0 (Djebali *et al*, 2012) non-default parameters), read counting on genes or exons (HTSeq, http://www-huber.embl.de/users/anders/HTSeq, mode: intersection-non-empty), and differential gene (DESeq2_1.4.5; Love *et al*, 2014) usage biostatistical analysis. PCA and distance heat maps were generated in R following instructions in the vignette for DESeq2. Genes were considered differentially expressed with an adjusted *P*-value lower than 0.05 (Benjamini-Hochberg). Gene set overlaps were calculated using Venny (http://bioinfogp.cnb.csic.es/tools/venny/). Pathway analysis was performed using Cytoscape (www.cytoscape.com) and Cytoscape extension ClueGO.

## Electrophysiology on hippocampal slices

### Slice preparation

Acute hippocampal slices prepared from WT mice and APPPS1ΔE9 mice were used for electrophysiological recordings. All the procedures were carried out in compliance to the guidelines from the Animal Committee on Ethics in the Care and Use of Laboratory Animals of TU-Braunschweig. Briefly, after anesthetization using $CO_2$, the mice were decapitated and the brains were quickly removed and cooled in 2–4°C artificial cerebrospinal fluid (aCSF). The hippocampi were dissected, and transverse hippocampal slices (400 μm) were prepared by using a manual tissue chopper. Then, slices were incubated for 3 h at 32°C in an interface chamber (Scientific System Design), which was continuously perfused with

oxygenated aCSF at a flow rate of 0.8 ml/min. The aCSF contained the following (in mM): 124 NaCl, 4.9 KCl, 1.2 $KH_2PO_4$, 2.0 $MgSO_4$, 2.0 $CaCl_2$, 24.6 $NaHCO_3$, 10 D-glucose, equilibrated with 95% $O_2$–5% $CO_2$ (32 L/h).

### Slice recordings

To evoke field EPSP (fEPSP) from Schaffer collateral/commissural-CA1 synapses, one monopolar lacquer-coated, stainless-steel electrode (5 MΩ; AM Systems) was positioned at the stratum radiatum layer of the CA1 region. For recording fEPSP (measured as its initial slope function), one electrode (5 MΩ; AM Systems) was placed in the CA1 apical dendritic layer and signals were amplified by a differential amplifier (Model 1700, AM Systems). The signals were digitized using a CED 1401 analog-to-digital converter (Cambridge Electronic Design). After the preincubation period, an input–output curve (afferent stimulation vs. fEPSP slope) was generated. Test stimulation intensity was adjusted to elicit fEPSP slope of ∼40% of the maximal fEPSP response. Four 0.2-Hz biphasic constant current pulses (0.1 ms/polarity) were used for baseline recording and testing at each time point. Long-term potentiation (LTP) was elicited by "strong" tetanus (STET) consisting of three stimulus trains of 100 pulses at 100 Hz delivered at 10-min intervals (300 pulses in total). Data (fEPSP slope) were normalized to baseline and plotted as average ± SEM. The average values of the slope function of the fEPSP (millivolts per milliseconds) for each time point were analyzed using paired *t*-test; $P < 0.05$ was considered as statistically significant.

### Thioflavin S staining

Thioflavin S staining was performed as previously described with the some modifications (Guntern *et al*, 1992; Sun *et al*, 2002). Briefly, mice were sacrificed and their brain quickly removed on ice. Left hemispheres were embedded in OCT (Tissue TEK) and stored at −80°C until ready to section. Sagittal, 25-μm-thick sections were cut using a cryostat. And, five sets of five slides per brain containing four sections/slide were made by taking every 5[th] section on the same slide, so that each set consisted of 20 representative sections throughout the brain. Sets to be used for thioflavin S staining were then fixed with 4% PFA for 25 min at room temperature, washed twice with 0.9% NaCl, and quickly rinsed with PBS. Staining was performed using filtered 0.05% thioflavin S in 50% ethanol for 8 min in the dark and differentiated with two changes of 80% ethanol for 10 s. This was followed by three washes with large volumes of distilled water and an incubation step in high-concentrated phosphate buffer (411 mM NaCl, 8.1 mM KCl, 30 mM $Na_2HPO_4$, 5.2 mM $KH_2PO_4$, pH 7.2) at 4°C for 30 min. Finally, slides were briefly rinsed in PBST and covered with coverslips using Vectashield Hard Set mounting media with DAPI (Vector). Slides were allowed to set in the dark at 4°C and imaged immediately thereafter. To control for background, unspecific staining, or tissue auto-fluorescence, slides from age-matched wild-type brains were used as negative controls. Images at 4× magnification of hippocampus and cortex were captured using an Olympus IX70 fluorescence microscope. Quantification of number of plaques, area covered by the plaque, and average plaque size was performed using the particle analysis tool of the ImageJ software. Statistical analysis was carried out using GraphPad Prism7.

## Biophysical experiments on Aβ$_{1-42}$

### Materials

Aβ$_{1–42}$ was purchased from Bachem and Anaspec (Fremont, CA).

The phospholipids 1.2-dioleoyl-*sn*-glycero-3-phosphoserine (DOPS) and 1-palmitoyl-2-oleoyl-*sn*-glycero-3-phosphoethanolamine (POPE) were purchased from Avanti Polar Lipids (Alabaster, AL). All other chemicals were purchased from Sigma-Aldrich (St. Louis, MO).

### Peptide handling

For BLM experiments, Aβ$_{1–42}$ peptides were dissolved in Milli-Q water to a concentration of 1 mg/ml prior to being aliquoted for storage. These 50-μl aliquots were stored at −80°C for a maximum of 60 days before use. Samples were thawed only once. For AFM experiments, powder form of Aβ$_{1-42}$ (Bachem, Torrance, CA) was first dissolved in 1% ammonium hydroxide until the peptides were completely dissolved. They were subsequently sonicated for approximately 2 min. Small volume of peptides were then aliquoted and lyophilized using a lyophilizer (Labconco FreeZone 2.5 Plus, Kansas City, KS). The aliquots were stored at −80°C until used. For every experiment, aliquoted peptides were thawed and hydrated in 20 mM NaOH and HEPES buffer solutions at pH 7.4, sequentially. The peptide concentration was measured using the 280 nm UV absorbance (ε = 1,490 $M^{-1}$ $cm^{-1}$).

### Peptide aggregation

Synthetic Aβ$_{1-40}$ and Aβ$_{1-42}$ powders were dissolved in 20 mM NaOH at 2 mg/ml concentration and incubated at 4°C for 1 h to dissolve their pre-existing aggregates. The Aβ stock solutions were then brought to PBS, pH 7.4, at a final peptide concentration of 0.4 mg/ml, around 90 μM. After a short centrifugation (16,000 *g*, 30 min), the supernatants were used either directly, as the monomeric Aβ, or following a 3-h incubation at 37°C (without agitation), as the oligomeric-enriched Aβ samples. The monomeric and oligomeric-enriched Aβ samples were then added to primary neuronal cultures for membrane integrity and viability assays.

### Electrical recording of planar lipid bilayer made of oxidized cholesterol/n-decane

We prepared planar lipid bilayers using the Müller-Rudin technique (Benz *et al*, 1978). The membranes were formed from a 1% (w/v) solution of oxidized cholesterol in *n*-decane. Oxidized cholesterol was prepared according to an established procedure (Benz *et al*, 1978). The lipid membranes were formed using a Teflon loop to spread the lipid solution across a hole with a surface area of approximately 0.3 $mm^2$ in the wall dividing the two aqueous compartments in a Teflon cell. After the membrane had turned black, the Aβ$_{1–42}$ peptide-containing protein fractions were added to the aqueous phase on one or both sides of the membrane at final concentrations 1 and 40 μM. For doping lipid bilayers with anle138b, the compound was dissolved in chloroform. The chloroform of a sample was then evaporated under vacuum. The lipid solution was added to the dry anle138b in such an amount that a final concentration of 1 mM in the lipid solution was obtained. This lipid solution was used for the normal protocol of bilayer formation as described above.

The current across the lipid bilayer membranes was measured with a pair of Ag/AgCl electrodes with salt bridges switched in series with a voltage source and a highly sensitive current amplifier Keithley 427 (Keithley Instruments, INC. Cleveland, OH). The output signal of the amplifier was recorded by a strip chart recorder (Rikadenki Electronics GmbH, Freiburg, Germany). The temperature was kept at 20°C throughout the experiment.

### Planar lipid bilayer electrical recording using DOPS and POPE

We prepared planar lipid bilayers using the so-called painted technique (Mueller *et al*, 1962). Anle138b was mixed with a 1:1 (w/w) solution of DOPS and POPE in chloroform at a concentration of 10 mM with respect to the volume of the lipids. A lipid specific gravity of 0.8 was used for the calculation. This mixture was subsequently dried in a Rotavapor R-210 (Buchi) and resuspended in decane at a total lipid concentration of 20 mg/ml. Bilayers with embedded anle138b were formed from this solution. Spontaneous membrane formation occurs following the addition of lipid directly over a pore with a diameter of ~250 μm in a Delrin septum (Warner Instruments, Delrin perfusion cup, volume 1 ml). In previous studies, this membrane composition was shown to be stable for long recording times (~4 h; Capone *et al*, 2012). Control experiments establishing the stability of membranes formed with the addition of anle138b were performed. We used 150 mM KCl, 10 mM HEPES (pH 7.4), and 1 mM $MgCl_2$ as the electrolyte.

We observed the following difficulty in the preparation of anle138b loaded lipids. Anle138b was dissolved in decane along with the lipids prior to membrane painting. Since anle138b is soluble in both the decane and the lipids, the distribution of compound in the lipid membrane that spontaneously forms upon lipid deposition over the aperture can vary. Lipid monolayers bind to either side of the partition and the bilayer membrane forms as the monolayers fuse together at the center, excluding the decane solvent to the perimeter. This solvent annulus acts as a bridge to the Delrin partition and is essential for membrane stability (White, 1972). If a significant proportion of the anle138b is mobile in the decane, the compound could be partitioned to the solvent annulus rather than incorporated into the membrane leading to BLM results that appear similar to that seen with $A\beta_{1-42}$ in the absence of compound. This can explain why anle138b modulated the activity of the pores in only 50% of the cases.

Before performing electrical recordings, we verified that the bilayer was stable for several minutes with low conductance (<10 pS) across ± 100 mV applied voltage and that the system capacitance was >110 pF. When both criteria were met, peptide was added directly to the *cis* (hot wire) side and stirred for 5 min. Peptide concentration in the bilayer chamber was approximately 10 μM. Bilayer stability was monitored by periodic capacitance measurements throughout the course of the experiment.

All traces were recorded in voltage clamp mode using the 2 kHz built-in filter cutoff of our BC-535 amplifier (Warner Instruments, Hamden, CT). A sampling frequency of 15 kHz was used for all data acquisition. We used a custom-made LabVIEW program to record the current and Clampfit 10.2 (Molecular Devices, Sunnyvale, CA) to analyze traces. We have filtered the recorded current versus time traces with a digital Gaussian low-pass filter and a cutoff frequency of 50 Hz for representation in figures.

### Lipid bilayer preparation for AFM imaging

For liposome preparation, DOPS and POPE lipids were used in a 1:1 ratio (Avanti Polar Lipids, Alabaster, Al). Liposomes were prepared by mixing 20 μl of each lipid (5 mg/ml) dissolved in chloroform, and anle138b, also in chloroform, was added to a 1,000:1 lipid to anle138b molar ratio. Then, liposomes were allowed to dry overnight in vacuum. The dried lipid film (and anle138b) was hydrated with peptide solution (1:60 peptide to lipid molar ratio) to facilitate peptide incorporation in the lipid bilayer, resulting in proteoliposome formation. For controls, the dried lipid film (and anle138b) was hydrated with 200 μl of HEPES buffer and vortexed occasionally for an hour. The large multilamellar vesicles formed with this procedure were sonicated for 5 min. Supported lipid bilayers were formed by (proteo)liposome rupture and fusion on the mica substrate (Lin *et al*, 2001; Quist *et al*, 2005; Liu *et al*, 2006; Umehara *et al*, 2010; Connelly *et al*, 2012a,b). Lipid concentrations of 0.1–1 mg/ml were deposited on freshly cleaved mica and incubated for ~10 min on a hot plate above the lipid transition temperature to facilitate fusion of the ruptured proteoliposomes on the mica surface. As a last step, samples were rinsed with buffer to remove unruptured proteoliposomes still in solution.

### AFM imaging on membranes

Topographic images were acquired using a Multimode AFM equipped with a Nanoscope V controller (Bruker, Santa Barbara, CA). Silicon nitride cantilevers with nominal spring constants of 0.08 N/m (OMCL-TR400, Olympus) were employed for imaging in fluid using the tapping mode. Resonance frequencies of ∼ 8 kHz and drive amplitudes under 100 mV were used. All experiments were performed at room temperature. Actual spring constants were measured to be within 10% of its nominal value using thermal tune before the experiments. To measure outer pore diameters, tip broadening was taken into account by modeling the inserted oligomers as a spherical cap protruding a height h above the surface of the lipid bilayer in contact with a spherical tip of radius R (Lee *et al*, 2014).

## Cell membrane integrity and viability

### CyQUANT

Primary neuronal cultures were produced from E17.5 CD1 Swiss embryos. On DIV, 10 cultures were treated conditioned medium supplemented with anle138b to a final concentration of 1 μM in 0.05% DMSO (Roth, A994.2) or 0.05% DMSO as vehicle. After 24 h $A\beta_{1-40}$ oligomers, monomers or buffer (*n* = 4 each) was applied at 10 μM and incubated for 48 h. CyQUANT® Direct Cell Proliferation Assay (Thermo Fisher, C35011) was used according to manufacturer's protocol to determine membrane integrity. After 30-min incubation, fluorescence was measured with a Tecan infinite 200. Statistical analysis was performed in GraphPad Prism.

### MTT assay

Cell viability was measured using the MTT assay with the same sample preparation as for the CyQUANT assay. Briefly, after anle138b and $A\beta_{1-40}$ treatment, the cell culture medium was supplemented with MTT to a final concentration of 0.5 mg/ml and incubated for 1 h at 37°C in a standard cell culture incubator. Subsequently, medium was removed and metabolites suspended in 500 μl DMSO. Absorption at 800 nm was measured using a Tecan infinite 200.

## Transgenic flies and survival assays

w[1118]; P{UAS-*Arctic* Aβ 1-42 (Glu22Gly)}(arc2E) transgenic flies were kindly provided by D. Crowther (Cambridge, UK; Crowther *et al*, 2005). The co-expression of Gal4 under a tissue specific promotor leads to activation of an upstream activating sequence (UAS) element that activates the transcription of Aβ42 upon Gal4-UAS binding (Brand & Perrimon, 1993). We controlled the temporal expression of the *Gal4*-UAS system by a temperature-sensitive Gal80ts under a ubiquitously expressing *tubulin* promoter (tub-Gal80ts) (McGuire *et al*, 2003). Gal80ts represses the transcriptional activity of *Gal4* at the permissive temperature (18°C). Upon a shift to the non-permissive temperature (30°C), Gal80ts becomes dysfunctional and releases the *Gal4*-UAS system to allow transgene expression. Neuron-specific and temperature-inducible expression of Aβ42 was achieved by crossing the w[1118]; Gal4-elav/CyO; TM2/TM6 and w[1118]; Sp/CyO; tubGal80ts/TM2 lines obtained from Bloomington (Bloomington *Drosophila* Stock Center (BDSC), Indiana University, USA) to generate w[1118]; Gal4-elav/(CyO); tubGal80ts/(TM6). The w[1118]; P{UAS-*Arctic* Aβ 1-42 (Glu22Gly)} line was crossed with w[1118]; Gal4-elav/(CyO);tubGal80 ts/(TM6). w[1118]; Gal4-elav/UAS-arc2E;tubGal80/+ were used for subsequent survival assays. As an overexpression control, we crossed w[1118]; Gal4-elav/(CyO);tubGal80/(TM6) with w[1118]; P{w[+mC]=UAS-lacZ.NZ}20b (Bloomington *Drosophila* Stock Center (BDSC), Indiana University, USA) to obtain w[1118]; Gal4-elav/+; tubGal80ts/UAS-lacZ.

## Survival assay

Male flies expressing Aβ42 Glu22Gly arctic mutation and control flies expressing *lacZ* were collected and maintained under a 12-h light–dark cycle in 60% relative humidity and equal population density per vial. Within 24 h after eclosion from the pupae, all flies were transferred to the restrictive temperature. Compounds were dissolved at 10 mM in DMSO and mixed with yeast. Fresh food was added every 2nd–3rd day. As a control, Aβ42 Glu22Gly transgenic and lacZ control flies were treated with DMSO food without addition of compounds. Surviving flies were counted every 2nd–3rd day. The resulting Kaplan–Meier survival curves were calculated using SSPS software. Differences between the genotypes were assessed by log-rank test (SSPS software).

## Statistical analysis

Unless specifically mentioned otherwise, data were analyzed by unpaired Student's *t*-test, two tailed *t*-test, Bonferroni test for multiple comparisons, or one and two-way and ANOVA (analysis of variance) when appropriate. Errors are displayed as standard error of mean (SEM). Unless otherwise stated, analysis was performed using GraphPad Prism.

## Data availability

RNA-sequencing datasets are available via GEO accession number: GSE104424.

**Expanded View** for this article is available online.

---

### The paper explained

**Problem**

A hallmark of Alzheimer's disease is the aggregation of Aβ, yet the mechanism of dysfunction of neurons and neuronal death is unclear and disease-modifying therapeutic intervention is not yet possible.

**Results**

Using the APPPS1 overexpression model in mice, we show that the diphenylpyrazole compound anle138b, an orally available small molecule reaching 30 μM concentration in the brain when given orally, ameliorated disease phenotypes such as hippocampal LTP, spatial reference memory, and transcriptional homeostasis. Biophysical analysis suggests that pore formation contributes to the dysfunctionalization and death of neurons, while anle138b prevents pore formation and thereby rescues these effects.

**Impact**

Our work provides evidence of the molecular mechanisms by which Aβ aggregation is contributing to Alzheimer's disease, namely pore formation in membranes by Aβ oligomers whose conductivity is reduced by more than a factor of 10 by anle138b treatment. The pore conductivity reduction correlates with the full refunctionalization as measured by LTP when anle138b is given before or after the onset of amyloid deposition. Also memory function is improved. Our data suggest that therapeutical effects can be expected to be achieved in AD patients with anle138b.

## Acknowledgements

We thank Lina-Maria Jaime Tobonand Inga Urban for assistance. This work was supported by core funds from the DZNE (to AF, MZ, and MF), the Max Planck Society (to GE and CG), the Hans and Ilse Breuer Award for Alzheimer's disease (to AF), the DFG project FI981/9-1 (to AF), the EU (ERC consolidator grant to AF), and the DFG (SFB803 project A04 to CG). HU holds a PhD fellowship from the Hans and Ilse Breuer Foundation. MF is supported by the CoEN Initiative CoEN3018 and the SFB1089 C01. NV is supported by the Malta Council for Science & Technology through the National Research & Innovation Programme (R&I-2008-068) and the University of Malta (PHBR06). AC was supported by the Malta Government Scholarship Scheme. J.L., A.L.G., and R.L. acknowledge support from the National Institute on Aging of National Institutes of Health (Grant AG028709).

## Author contributions

AHM: coordinated the experiments, performed behavior testing of mice, RNA-seq, and amyloid plaque analysis. HU: performed all experiments related to primary hippocampal neurons. ALG: performed the electrophysiology with black membranes. JL: did AFM measurements of Aβ on membranes. SR: provided anle138b. HYA: performed behavior experiments. GJ: analyzed RNA-seq data. EB: performed pathway analysis. GG: performed the LTP measurements. AL: provided anle138b. NR-G: generated synthetic Aβ used for the CyQuant and MTT assay. FTA: did the AFM measurements. PW: performed fly experiments. AS: designed and analyzed fly experiments. MZ: generated synthetic Aβ, wrote the manuscript. RB: performed the BLM measurement. AG: discussed the data. MK: conceived and supervised experiments, discussed and analyzed specifically the electrophysiological experiments, wrote the manuscript. RL: conceived and supervised experiments, specifically supervised the AFM measurements and interpreted them, wrote the manuscript. CG: conceived and supervised experiments, wrote the manuscript. GE: conceived and supervised experiments, wrote the manuscript. AF: conceived and supervised experiments, wrote the manuscript. LK: helped to perform and analyze

RNA-seq data. JW and MF treated Tau mice with anle138b and collected tissue. MC, AC, and NV performed and interpreted the experiments on cytochrome C release from mitochondria.

## Conflict of interest

AG and CG are co-founders of MODAG. AL is partly employed by MODAG.

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
