## [Review Process File · EMBO Molecular Medicine]

The diphenylpyrazol compound anle138b blocks A β channels and rescues disease phenotypes in a mouse model for amyloid pathology.

Ana Martinez Hernandez, Hendrik Urbanke, Alan L. Gillman, Joon Lee, Sergey Ryazanov, Hope Y. Agbemeyah, Eva Benito, Gaurav Jain, Lalit Kaurani, Gayane Grigoryan, Andrei Leonov, Nasrollah Rezaei-Ghaleh, Petra Wilken, Fernando Teran Arce, Jens Wagner, Martin Fuhrman, Markus Zweckstetter, Roland Benz, Armin Giese, Anja Schneider, Martin Korte, Ratnesh Lal, Christian Griesinger, Gregor Eichele and André Fischer

Corresponding authors: André Fischer, University Medical Center Goettingen; Martin Korte, Technical University Braunschweig; Ratnesh Lal, University of California San Diego; Christian Griesinger, Max Planck Institute for Biophysical Chemistry; Gregor Eichele, Max Planck Institute for Biophysical Chemistry

Review timeline:

Submission date:	23 March 2017
Editorial Decision:	24 April 2017
Revision received:	28 August 2017
Editorial Decision:	15 September 2017
Revision received:	17 October 2017
Accepted:	20 October 2017

Transaction Report:

Editor: Céline Carret

1st Editorial Decision

24 April 2017

Thank you for the submission of your manuscript to EMBO Molecular Medicine. We have now heard back from the two referees whom we asked to evaluate your manuscript.

You will see from the comments below that both referees have overlapping concerns on the paper, namely limited novelty, mechanism and in vivo relevance. Both reports are nicely detailed and good suggestions are provided to perform additional set of experiments/analyses to make the conclusions stronger. Explanations and clarifications are also needed as some discrepancies are found. As the anle138b compound was reported before to have some beneficial effects on neurodegenerative diseases, further cross-commenting revealed one way to address the limited advance of the findings, and this could be by performing some sort of "dose-response curve in the different models (PrP, Htt, Tau, Abeta) to determine which of the different pathologies would be more susceptible to respond to this type of treatment. This addendum, along with some of suggested experiments would definitely increase mechanistic insights and in vivo biological significance and would be then make the paper better suited for EMBO Molecular Medicine.

I look forward to receiving your revised manuscript.

***** Reviewer's comments *****

Referee #1 (Comments on Novelty/Model System):

The work is technically correct, the conclusions according to the data and of potential clinical relevance: authors test the effect (at the electrophysiol and behavioral level) of a compound that interferes with amyloid oligomers-induced membrane pores in a transgenic mouse model of AD (model is not great but there aren't good models for this disease). I have however three main concerns: i) novelty (this compound ameliorates the effect of other protein aggregates, namely PrPc, tau and synuclein), ii) the mechanism of action (authors propose that is by inducing a conformation change in the amyloid pore but it could also be prevention of oligomerization) and iii) rather modest phenotypic characterization of the phenotype of mice exposed to treatment.

Referee #1 (Remarks):

In this manuscript, Hernandez and colleagues describe the beneficial effect of the of the compound anle138b in a mouse model (transgenic) of AD: oral administration of the compound rescues LTP and to a certain (moderate) extent a spatial memory defect of these mice. Importantly, oral treatment rescued the gene expression changes induced by the double transgene of these mice in the early stages of the pathology but not those in the late stage, especially those related to inflammation, suggesting that the synaptic plasticity rescue is at a different cellular/molecular level. Authors hypothesize that this might be in the plaque forming process and in fact, treatment reduces the number of plaques in both early and late phases of disease. In addition, the studies in artificial membranes suggest that anle138b makes A β 1-42 conducting pores to non-conducting, making authors consider this another mechanism by which anle138b rescues reduced LTP and learning deficits in APPPS1 Δ E9mice. Work in cultured hippocampal neurons also revealed that abeta oligomers induce membrane damage and that this effect is prevented by incubating cells with this compound. Although the data in this last aspect are convincing, I am not sure the result proves that abeta makes conducting pores in living cells, rather than inducing membrane damage by a different mechanism (i.e. sequestering lipids) which is then prevented due to fibril binding capacity of the compound (like on synuclin fibrils).

The paper is well written, in a very clear and concise manner. The data are solid, the conclusions supported by the data and the potential implications for the treatment of this disease evident. On the other hand, I have a series of concerns that diminish my enthusiasm for its acceptance at EMBO Mol. Med. One of them is novelty: this compound not only has been demonstrated effective against α -synuclein and prion toxicity (as authors mention), but also in tauopathies (see "Reducing tau aggregates with anle138b delays disease progression in a mouse model of tauopathies" Wagner J, et al. Acta Neuropathol. 2015, 130(5):619-31) and in a Parkinson's mouse model (see "The oligomer modulator anle138b inhibits disease progression in a Parkinson mouse model even with treatment started after disease onset", Levin J, et al. Acta Neuropathol. 2014, 127(5):779-80). Moreover, structural studies already showed that this compound possesses high binding affinity to protein aggregates, using monomeric and aggregated α -synuclein for the study (Deeg et al., 2015). Hence, a second concern is on the mechanism. Authors on the one hand report a reduced plaque load in treated mice and on the other pore forming-membrane damaging prevention. In light of the Deeg paper, this compound may induce changes in aggregation of abeta oligomers, rendering them to a monomeric state. This could be tested. A third concern, linked to the previous, is about the existence of abeta pores in vivo. The data in artificial membranes clearly helps to accept this view, though the data in neurons is not. Again, membrane integrity would occur in the oligomers become monomers. Finally, because of the rather protein aggregation unspecific effect of this compound (precludes toxicity of prion, synuclein, tau) authors may want to: i) perform a more comprehensive study of treated mice phenotype (by histology, other behavioural traits, lifespan length), and ii) study the effect of this compound in neurons with another type of beta sheet, amyloid-like aggregates (mutant huntingtin).

Referee #2 (Comments on Novelty/Model System):

The manuscript is another report of the series of the papers published by the authors on the potential application of anle138b for the treatment of neurodegenerative disease. The authors use a combination of cutting-edge molecular biology approaches to investigate the molecular mechanisms of the anle138b action. However, the model system they applied in parts of the manuscript uses very high amounts of recombinant amyloid beta, and hence may not recapitulate the physiological settings.

Referee #2 (Remarks):

The manuscript submitted by Martinez Hernandez et al. provides another report of the series published by the group on the use of the anle138b oligomer modulator for the treatment of neurodegenerative diseases. The study reports a novel molecular mechanism, via which anle138b may halt Alzheimer's disease (AD) progression. The findings of the potential for anle138b to rescue behavioural and LTP impairments and to prevent amyloid deposition in the APPPS1 Δ E9 mice are interesting. However, additional experiments are required to determine the physiological/in vivo relevance of the molecular mechanisms of the anle138b effect on the A β channels. The following points should be addressed.

Major comments:

1. Additional control, the wt + placebo animals, should be included in the behavioural studies as well, to support the claim that anle138b action is mediated through the interference with A β overload and to rule out the possibility that anle138b does not simply affects the behaviour through other mechanisms even in wild type mice, presenting relatively low endogenous A β levels.
2. The findings presented in the text, figure 3 and supplementary table 1 are very confusing and inconsistent. In the section describing the transcriptome analysis, the authors report in the text that 202 genes were differentially expressed between wt and APPPS1 Δ E9 placebo-treated mice and this number was reduced to 27 genes when wt-anle138b vs APPPS1 Δ E9-anle138b groups were compared. There is discrepancy between this statement and the data presented in the graph in Figure 3A, which shows the opposite. Further discrepancy is apparent in the supplementary table 1. In the second and fourth section of the table the authors present the genes differentially expressed between placebo and anle138b treated APPPS1 Δ E9 mice, and show that APP and PS1 are upregulated, which is opposite to the text and the graph in figure 3C. The authors need to reconcile the information in the text, figures and tables. Otherwise it is impossible to understand and interpret the data, and thus they become unreliable. Moreover, the transcriptomics data should be validated using a complementary approach, such as immunostaining or western blotting, to demonstrate that these alterations correspond to the changes in the expression of respective proteins. This is especially important for the data reporting the lack of reduction (according to the text) in inflammatory response, a well-established hallmark of AD pathology, in the anle138b-treated animals. In contrast, in the table authors list the inflammatory genes as changed between placebo vs. anle138b treated APPPS1 Δ E9 mice in the post-plaque group. Do clarify whether they were changed or not?
3. In Figure 4 the authors present the data on the amyloid pathology. Representative images for all the conditions should be presented. Although there is a reduction in the amyloid pathology in the anle138b-treated post-plaque mice, these animals still have much more amyloid burden than the placebo-treated pre-plaque animals. Since the latter present severe behavioural/LTP abnormalities, then one would expect that the anle138b-treated post-plaque animals would present severe impairments as well, if the amyloid deposition was causative of those. However, according to the data presented in Fig.2, they resemble wild type animals. This should be explained. Moreover, in vitro assays demonstrating the effect of the anle138b on the amyloid oligomerization/aggregation should be performed.
4. The control data in the supplementary figure 3 present lack of anle138b effect on the thickness of the lipid bilayers. Does A β treatment affect the thickness? Is this potential A β effect rescued by anle138b? These additional controls should be included.

5. The AFM imaging of the A β pores demonstrates lack of changes in the channels structure. Was the number of channels formed altered in the anle138b vs vehicle-treated membranes? Quantitative analysis should be performed.

6. In figure 5, the authors present the findings from primary neurons. The concentration of A β is very high in these experiments (10 μ M), largely exceeding the physiological amount, especially at the pre-plaque stage. To provide physiological relevance and to ensure that the effect observed in the cultured neurons is not an artefact of the very high A β concentration either more physiological dose of recombinant A β or naturally secreted A β in conditioned medium should be used.

Please provide details of the preparation and validation of A β monomers and oligomers used.

In the method the authors state that these were A β 40 oligomers while in the figure legend and the text that A β 42 was used. This information needs to be clarified.

The authors demonstrate that the membrane integrity is better preserved in the anle138b pre-treated neurons when compared to the vehicle pre-treated cells, suggestive that anle138b may prevent formation of the A β pores rather than alterations in their kinetics. Reverse treatment, i.e. pre-treatment with A β followed by addition of anle138b, should be performed to determine if anle138b affects the pore kinetics.

Minor comments:

1. Only figure 1 has a number included in the image.
2. The authors describe hippocampal gene expression data on page 6, last paragraph, and refer to figure S1. However, this figure presents data in flies.
3. Using consistent formatting (colors and decimal places) for different sections in the supplementary table for the pre- and post-plaque group would be helpful.
4. Please rewrite the section on the formation/dynamic/conductance of the A β pores in the lipid bilayers (page 8, second paragraph), as the current version is somewhat confusing.
5. Supplementary figure 3 contains data on the DOPC/DPPC lipid bilayers, which were not used through the manuscript and are not described in the methods. Please refer to the data, explaining why they are relevant or remove it from the manuscript.

1st Revision - authors' response

28 August 2017

While referee 1 says that our "...*paper is well written, in a very clear and concise manner. The data are solid, the conclusions supported by the data and the potential implications for the treatment of this disease evident.*", he/she raises a number of specific questions.

Referee #1: point 1:

He/she states "*I am not sure the result proves that abeta makes conducting pores in living cells, rather than inducing membrane damage by a different mechanism (i.e. sequestering lipids) which is then prevented due to fibril binding capacity of the compound (like on synuclin fibrils).*"

We appreciate this insightful comment. We also agree that our data provides evidence, but does not definitely prove the existence of pores *in vivo*. For this reason we had included the *in vitro* experiments to study to role of A β and anle138b on pore formation in lipid bilayers. Thus, taken together the *in vivo* and *in vitro* data provide a compelling argument that at least part of the observed pathology and therapeutic effect of anle138b is mediated via this mechanism. We discuss this issue now in greater detail, provide additional references to support our interpretation and specifically address this reviewers concern in that we acknowledge that our data strongly argues that the proposed mechanisms plays a role, but that other processes also may contribute to the observed effects. See page 3, lines 10-13, page 11, lines 28-31; page 12, lines 30-33; page 13, lines 1-15 and 21-33 , page 14, lines 1-2 and 32-33, page 15, lines 10-18 of the revised manuscript

Please also see response to referee #2, point 7, since in response to the concern raised there we have conducted additional experiments on membrane integrity.

Referee #1: point 2:

He/she comments on the novelty of our data and says: “...*this compound not only has been demonstrated effective against I±-synuclein and prion toxicity (as authors mention), but also in tauopathies (see "Reducing tau aggregates with anle138b delays disease progression in a mouse model of tauopathies" Wagner J, et al. Acta Neuropathol. 2015, 130(5):619-31) and in a Parkinson's mouse model (see "The oligomer modulator anle138b inhibits disease progression in a Parkinson mouse model even with treatment started after disease onset", Levin J, et al. Acta Neuropathol. 2014, 127(5):779-80).*”

We appreciate this comment and apologize that our data was presented obviously not in an optimal manner. We understand that this referee challenges the novelty of our study, on the basis that a therapeutic effect of anle138b have been reported for other neurodegenerative diseases.

We now clarify this issue. First, we like to state that anle138b has not been tested in a mouse model for A β -pathology. As such, all of the data presented in our manuscript are novel. We also like to mention that just because a small molecule compound was found to be effective for prion, α -Synuclein and more importantly also for Tau pathology it does not necessarily have to be effective in a mouse model for A β pathology. The finding that anle138b ameliorates disease phenotypes in a mouse model for Tau and A β pathology is thus very exciting. Taking into account that Tau and A β pathology represent the two major hallmarks causatively linked to AD pathogenesis, we would argue that our data is of utmost importance. In fact, most of the therapeutic strategies developed in the past either aim to affect Tau or A β pathology or aim for symptomatic treatments related processes such as inflammation or gene-expression.

To find a small molecular that would causatively affect the two major hallmarks of AD, is even effective after the onset of the disease and has excellent blood/brain permeability is thus a major aim in the field of AD research.

We discuss these issues now in greater detail in the revised manuscript on page 8, lines 1-27; page 11, lines 13-14; page 12, lines 30-34; page 13, lines 1-15 and page 15, lines 26-30; .

Moreover, to strengthen the argument that anle138b ameliorates A β and Tau phenotypes we teamed up with the Fuhrmann-group that demonstrated the therapeutic effect of anle138b in TauP301S mice but did not analyze – as we did in our study – the hippocampal transcriptome as an estimate of brain homeostasis. By performing RNA-seq from the hippocampus of the same TAUP301S mice used in the experiments described by Wagner et al., 2015 (we were able to obtain hippocampal tissue from the Fuhrmann group), we find that Tau pathology correlates with a massive change in hippocampal gene-expression, which is partially ameliorated after anle138b treatment. These data are in line with our findings in APPPS1 Δ E9 and is now shown as novel Fig S4 and discussed in the revised manuscript on page 8, lines 1-27, page 12, lines 30-34 and page 13, lines 1-15

Referee #1: point 3:

He/she comments: “*Moreover, structural studies already showed that this compound possesses high binding affinity to protein aggregates, using monomeric and aggregated I±-synuclein for the study (Deeg et al., 2015). Hence, a second concern is on the mechanism. Authors on the one hand report a reduced plaque load in treated mice and on the other pore forming-membrane damaging prevention. In light of the Deeg paper, this compound may induce changes in aggregation of abeta oligomers, rendering them to a monomeric state. This could be tested.*”

We are thankful for this insightful comment. This reviewer points nicely to the Deeg et al. paper where indeed binding of anle 138b to aggregates of α -synuclein could be shown. Similar binding was seen for tau paper (Wagner et al. 2015). In addition, in the Wagner et al 2013 paper it was described that *in vivo* the larger oligomers of α -synuclein (lane 3 and 4 in Fig. 8f) were reduced upon treatment, similarly the larger oligomers of tau (Fig. 3e-h) were reduced. Since the larger oligomers

of these proteins are on-pathway aggregates to the fibrils while the compound bound aggregates are not, fibrils are generally reduced in these mouse models. Since the oligomers also make pores while with anle138b they don't, the inhibition of pore formation and the reduction of aggregates are not different mechanisms but rather two sides of the same coin. We did not look at the A β oligomer distribution in the same way as for tau and α -synuclein because of the small size of A β which would have made UZ unpractical.

We made several attempts to follow the aggregation inhibition of A β in the presence of vesicles with and without anle138b. We had technical problems and therefore are not able to show the data here. We believe that this is because such experiments do not reflect the situation found *in vivo*, most probably due to the fact that cytosolic proteins have the ability to dissolve anle138b at higher concentrations. We performed experiments with a more soluble variant of anle138b, namely anle138c. In that case, no membranes are necessary. Indeed, then aggregation is inhibited according to ThioT and also EM, but also pore formation is stopped. We refrained from putting these experiments into the manuscript, since they refer to a compound that is much more hydrophilic (solubility approx. 200 times higher in water than for anle138b) and since this compound is useless in animal models because of extremely fast metabolism in the liver. However, we like to share this data with the referee and otherwise would like to leave it up to his/her and the editors decision if this data should become part of the appendix. In any case, since the review process will be published the data will be available to the expert interested in this data (See figure below).

Figure. a) Thioflavin T fluorescence data showing the inhibitory effect of anle138c on A β ₁₋₄₂

aggregation (10 μ M, black) for 2 concentrations of compound, 1 μ M (red) and 10 μ M (blue). Data averaged over 3 data sets. b) and c) AFM images demonstrating a significant reduction in the number of fibrils for 100 μ M $A\beta_{1-42}$ solutions incubated for 3 days at 37 °C in the absence of compound b), and with anle138c compound c). The compound to peptide ratio was 1:1, as in a). d) Temporal variation of Thioflavin T (ThT) fluorescence intensity, as a probe of $A\beta_{1-42}$ amyloid fibril formation. $A\beta_{1-42}$ and anle138c concentrations were 50 and 60 μ M, respectively. The compound anle138c remarkably inhibits formation of ThT-reactive fibrils similar to the inhibition of $A\beta_{1-42}$ aggregation. e) and f) show electron micrographs after the aggregation in the absence (e) and presence of anle138c (f) which show that anle138c stabilizes oligomers rather than monomers

Referee 2 points out that the anle138b treated post-plaque group has more aggregates than the untreated pre-plaque group and asks the question how this can be reconciled. This is a nice argument to further consolidate the view that anle138b's activity is on the oligomers (less toxic oligomers in the treated post-plaques group than in the untreated pre-plaque group) rather than the fibrils (less in the untreated preplaque group than in the treated post-plaques group). See also response to point 4 of referee 2

Moreover, the question asked here would also be difficult to be address with biophysical experiments. As the AFM experiment in which Abeta was added to membranes doped with anle138b anle138b does not prevent the formation of membrane associated structures and does not change their macroscopic appearance. The distinction of the two mechanisms therefore is indeed difficult, since the compound stabilized oligomers produce less fibrils. Thus even if we did the proposed experiment it would not allow to distinguish between the different mechanisms

Referee #1: point 4:

He/she says: "A third concern, linked to the previous, is about the existence of abeta pores *in vivo*. The data in artificial membranes clearly helps to accept this view, though the data in neurons is not. Again, membrane integrity would occur in the oligomers become monomers."

This referee reiterates his question put forward previously (See referee 1, point 1) regarding the existence of $A\beta$ pores *in vivo*. Indeed, it is very difficult to demonstrate $A\beta_b$ -pores *in vivo* e.g. by imaging. We are very thankful that referee 1 appreciates the impact of our *in vitro* experiments using lipid bilayers. We also agree that the *in vivo* experiments in neurons support an effect of anle138b on $A\beta$ pores but do not exclude other explanations. Indeed, if the oligomers became monomers, the membrane integrity results could be explained as well. However, a oligomer to monomer conversion would have involve other components than present *in vitro*, since to the best of what AFM can do, the number of oligomers in the membrane do not change for anle138b being absent or present. Yet, since for technical reasons this experiment cannot be done *in vivo*, we are afraid that we cannot give a final answer to this question. We would like to stress that the present state of art regarding the toxicity mechanisms and mechanisms for detoxification have not provided in any publication an answer to the question which process one has to block or which aggregate one has to remove. Our mechanistic insight however is, that pores are blocked by anle138b and animals treated with anle138b have better LTP and better memory. This correlation, however, is not there with the fibrillary aggregates. Whether the oligomers become non-conductive or become monomeric cannot be resolved *in vivo*. Yet, the *in-vitro* evidence is that the conversion to monomers is not a dominant process. Also from a thermodynamic view this is would be surprising. Anle138b has to bind to something. It could bind to monomers and stabilize them such that no oligomers and fibrils are formed. Yet, from various experiments, including the Deeg et al. data, we find no binding to the monomer. Binding to the pore forming oligomers can also not be the dominant process. Thus, the most reasonable (and also found for the above mentioned anle138c) is the stabilization of a different form of oligomers which are structurally different from the toxic ones and also exhibit different electrophysiological properties. In summary, while it would be great to see *in vivo* what we see *in vitro*, we hope that the reviewers don't insist on such experiments which have not been done in the literature. We can only state, that *in vitro* the mechanism is not oligomer to monomer conversion and can only say *in vivo*, that the oligomers are less toxic and less pore forming

Please refer to our answer to referee 1, point 3 for more details.

Referee #1, point 5:

This referee says: “*Finally, because of the rather protein aggregation unspecific effect of this compound (precludes toxicity of prion, synuclein, tau) authors may want to: i) perform a more comprehensive study of treated mice phenotype (by histology, other behavioural traits, lifespan length), and ii) study the effect of this compound in neurons with another type of beta sheet, amyloid-like aggregates (mutant huntingtin).*”

We understand that this comment is essentially repeating the concern raised by this reviewer in point 2. Therefore, please see our response to point 2.

In brief, to study the effect of anle138b in a mouse model for Huntington’s disease is certainly an interesting approach, that is however unrelated and beyond the scope our current study. Moreover, a more comprehensive analysis of the employed APPPS1ΔE9 mice would also not help to add more insight. For example life span length analysis is not only very time consuming but also not a *bona fide* phenotype to assay pathology in APPPS1ΔE9 mice. Rather, with the analysis of hippocampal LTP, spatial reference memory, explorative behavior and basal anxiety as well as hippocampal transcriptional plasticity we already provide a solid phenotypic analysis.

In addition, we addressed this concern experimentally in two ways

One main message to be communicated by our study, is that anle138b represents a suitable therapeutic approach to target the two major causative hallmarks of AD, namely amyloid and Tau-pathology. To strengthen this part of our study, we have now included novel data showing that anle138b treatment in TAUP301S mice ameliorates also hippocampal gene-expression changes (novel Fig S4). We discuss these findings in the revised manuscript on page 8, lines 1-27, page 12, lines 30-34 and page 13, lines 1-15.

Moreover, we have now also tested the effect of anle 138b on mitochondrial integrity (as measured by the Cytochrome release (CCR) assay that is indicative of mitochondrial membrane integrity) in response to Tau, a-synuclein and Aβ. This data is shown as novel supplemental figure 7B and discussed in the text on page 10, lines 25-32.

Referee #2

Referee #2, point 1:

Referee #2 says that we “*use a combination of cutting-edge molecular biology approaches to investigate the molecular mechanisms of the anle138b action. However, the model system they applied in parts of the manuscript uses very high amounts of recombinant amyloid beta, and hence may not recapitulate the physiological settings.*”

We completely agree with this comment. However, this argument is essentially true for all of the research in the field of AD and other neurodegenerative diseases. As such, the APPPS1ΔE9 of course only recapitulate part of the pathology seen in AD patients. Moreover, different levels and species of Aβ fragments and oligomers are observed in the various APP mouse models. To this end, the results obtained from such animal models have to be interpreted with great care. We now specifically refer to this issue on page 3, lines 28-31 and page 13, lines 10-12 of the revised manuscript.

This referee also specifically refers to the experiments in primary neurons that have been treated with Aβ. We are very aware of the fact that this system does not recapitulate the human disease. Moreover, various preparations and concentrations of Aβ peptides have been reported to have detrimental effects in cultures cells. Here the concentrations range from 0,5nM – 100 μM (e.g. see Cantara et al., 2005; Chen et al., 2006; Chafekar et al., 2008; Sondag et al., 2009; Choi et al., 2013; Lee et al., 2017). Thus, the concentration used in our study (10 μM) is well within the range of published data. The aim of the experiments in neuronal cells was to support the *in vitro* data on Aβ pores. Thus, we had to choose a concentration in which the employed Aβ preparation would not affect cellular function in the MTT assays but have an impact on membrane integrity. In this context

we could then test the effect anle138b. Moreover, to precisely define a “physiological” concentration of A β in cell culture experiments is very difficult since many aspects such as concentration near amyloid plaques would need to be taken into consideration and moreover a 2D-cell culture can never resemble the 3D-architecture of the brain.

See page 10, lines 15-20 of the revised manuscript. Moreover, as pointed out by this referee in point #7 there were some errors in the previous version of the text referring sometimes to A β ₁₋₄₀ and A β ₁₋₄₂, the latter one be more neurotoxic. We now corrected these mistakes and now state that in all experiments related to membrane integrity we employed A β ₁₋₄₀ (See also response to point 7, referee 2).

Referee #2, point 2:

He/she says: *“Additional control, the wt + placebo animals, should be included in the behavioural studies as well, to support the claim that anle138b action is mediated through the interference with A β overload and to rule out the possibility that anle138b does not simply affects the behaviour through other mechanisms even in wild type mice, presenting relatively low endogenous A β levels.”*

The behavioral data shown in Fig 1 & 2 indicate that anle138 does not affect memory function in wild type mice. However, we now evaluated the additional experiment and provide data that anle138b does not affect spatial reference memory when directly compared to wild type mice. This data is shown as novel supplemental figure 2 and discussed on page 5, lines 33-34 and page 6, line 1.

Referee #2, point 3:

“The findings presented in the text, figure 3 and supplementary table 1 are very confusing and inconsistent. In the section describing the transcriptome analysis, the authors report in the text that 202 genes were differentially expressed between wt and APPPS1 Δ E9 placebo-treated mice and this number was reduced to 27 genes when wt-anle138b vs APPPS1 Δ E9-anle138b groups were compared. There is discrepancy between this statement and the data presented in the graph in Figure 3A, which shows the opposite. Further discrepancy is apparent in the supplementary table 1. In the second and forth section of the table the authors present the genes differentially expressed between placebo and anle138 treated APPPS1 Δ E9 mice, and show that APP and PS1 are upregulated, which is opposite to the text and the graph in figure 3C. The authors need to reconcile the information in the text, figures and tables. Otherwise it is impossible to understand and interpret the data, and thus they become unreliable.”

We apologize for these mistakes.

1.

We realized that the labels in the lower panel of Fig 3A had been switched. This has now been corrected. Otherwise all numbers are correct.

2.

Regarding the second issue, the labeling of the tables in the supplemental data was indeed wrong due to a copy and paste error. We are really thankful that this referee took genuine interest in our work and spotted this mistake. The description in the main text was however correct as there we stated that we compared WT-anle138b vs. APPPS1 Δ E9-anle138b mice. Therefore, it is correct that APP and PS1 are increased in APPPS1 Δ E9-anle138b mice since they were compared to the corresponding anle138b-treated wild type groups. We now corrected these mistakes in the supplemental table and rewrote the text to avoid any misunderstanding. Please see underlined text on pages page 6 & 7. of the revised manuscript and the revised supplemental table 1.

Referee #2, point 4:

“Moreover, the transcriptomics data should be validated using a complementary approach, such as immunostaining or western blotting, to demonstrate that these alterations correspond to the changes

in the expression of respective proteins. This is especially important for the data reporting the lack of reduction (according to the text) in inflammatory response, a well-established hallmark of AD pathology, in the anle138b-treated animals. In contrast, in the table authors list the inflammatory genes as changed between placebo vs. anle138b treated APPPS1 Δ E9 mice in the post-plaque group. Do clarify whether they were changed or not?"

Previous studies from our group have demonstrated that RNA-seq data correlates well with qPCR analysis that is the gold standard to confirm sequencing results (e.g. See Peleg, et al., Science, 2010 or Benito et al. Journal of Clinical Investigation, 2015). We agree, however, that a confirmation is important.

Thus, we now check the RNA-seq data using qPCR. To this end we randomly selected in total 7 genes from the pre and post-plaque group that were significantly increased when we compared RNA-seq data from the placebo-treated wild-type vs. the placebo-treated APPPS1 Δ E9 mice. Our qPCR analysis shows that the RNA-seq and qPCR data significantly correlate (See novel Fig S3A).

For 3 selected genes linked to inflammation we also confirmed that their expression in the post-plaque group is still increased even after anle138b treatment. This data is now presented as novel FigS3B-and described on page 6 and 7 of the revised manuscript.

We could not provide data regarding the protein levels since we did not have protein lysates available and had used all brain section for the detection of amyloid plaques. Time restriction and the fact that our animal protocol for the experiments here ended, did not allow us to repeat all of the presented experiments in order to obtain tissue for protein analysis. However, since the aim of our study was to use gene-expression as a read out form hippocampal homeostasis, rather than to identify proteins that would help to explain the phenotype in APPPS1 Δ E9, we would appreciate if this referee would accept the provided qPCR data as sufficient to confirm the RNA-seq results.

Regarding the supplemental table, the confusion is due to the same issue addressed in response to point #2 raised by this reviewer. We now corrected the labeling of the table (See response to reviewer #2, point 2). To this end, inflammatory genes are indeed increased when comparing WT-anle138b vs. APPPS1 Δ E9-anle138b treated mice of the post-plaque group.

Referee #2, point 5:

"In Figure 4 the authors present the data on the amyloid pathology. Representative images for all the conditions should be presented."

We now include representative images as novel panel C in the revised Fig. 4.

Referee # 2 continues *"Although there is a reduction in the amyloid pathology in the anle138-treated post-plaque mice, these animals still have much more amyloid burden than the placebo-treated pre-plaque animals. Since the latter present severe behavioural/LTP abnormalities, then one would expect that the anle138-treated post-plaque animals would present severe impairments as well, if the amyloid deposition was causative of those. However, according to the data presented in Fig.2, they resemble wild type animals. This should be explained. Moreover, in vitro assays demonstrating the effect of the anle138 on the amyloid oligomerization/aggregation should be performed"*.

This is true and also expected since the post-plaque group presents a much more advanced stage of amyloid pathology. Thus, even after anle138b treatment the amyloid plaque load in the post-plaque group is much higher than the plaque load in the placebo-treated pre-plaque group that shows memory and LTP impairments.

These data are interesting for a number of reasons and indeed support our interpretation of the presented findings that anle 138b affect a toxic function of an A β oligomer for two main reasons:

- That restoration of memory function and LTP is observed in APP mice although amyloid plaque load was not altered is not uncommon (e.g. see Govindarajan et al.,

2013 or Benito, 2015) and is in line with the view that the amyloid plaques cannot be the sole reason for memory impairment in APP mice (Selkoe & Hardy, 2016; PMID:27025652)

- The finding supports our interpretation of the data that anle138b's activity is on the oligomers (less toxic oligomers in the treated post-plaques group than in the untreated pre-plaque group) rather than the fibrils (less in the untreated preplaque group than in the treated post-plaques group) as pointed out in the answer #3 to reviewer 1 who had a related question.

We now discuss this issue in greater detail. In summary, these data support our interpretation that an important mechanism by which anle138b ameliorates LTP and memory impairment is not the reduction of amyloid plaques but rather the restoration of membrane integrity. Please see page 13, lines 21-32, of the revised manuscript.

Referee #2, point 6:

He/she says *“The control data in the supplementary figure 3 present lack of anle138b effect on the thickness of the lipid bilayers. Does $A\beta$ treatment affect the thickness? Is this potential $A\beta$ effect rescued by anle138b? These additional controls should be included.”*

This is a very insightful comment. We had performed all of these control experiments but only included the data to demonstrate that the thickness of lipid membranes does not change in the presence of anle138b. Now, we also added data to show that $A\beta$ does not affect the thickness of the membrane, .. We include the corresponding AFM images of DOPS/POPE with or without $A\beta$ as novel panels in Fig S6, that was formally Fig S3. Please note that from former Fig S3 (now FigS6) we have now removed two panels showing data on DOPC/DPPC membranes, as suggested by reviewer 2 (see minor issue #5, referee 2). In addition we discuss the novel data on page 9, line 20 of the revised manuscript

Referee #2, point 7:

He/she states *“The AFM imaging of the $A\beta$ pores demonstrates lack of changes in the channels structure. Was the number of channels formed altered in the anle138b vs vehicle-treated membranes? Quantitative analysis should be performed.”*

Although we did not observe noticeable changes in $A\beta$ channel structures, the insertion of oligomers of $A\beta$ into the membrane and formation of $A\beta$ pores were observed. (See EV Fig 2). In the presence of anle138b they formed mainly non-conductive channels while in the absence the channels were conductive. Since AFM is not an integral method and focusses only on a small area, a bulk quantization of conductive and non-conductive channels is difficult to obtain from AFM. Yet, we can clearly say, that anle138b does not prevent $A\beta$ oligomers/pores from inserting in the membrane. Thus the mechanism that anle138b would dissolve oligomers to monomers is clearly in disagreement with the AFM images obtained.

Referee #2, point 8:

“In figure 5, the authors present the findings from primary neurons. The concentration of $A\beta$ is very high in these experiments (10 μ M), largely exceeding the physiological amount, especially at the pre-plaque stage. To provide physiological relevance and to ensure that the effect observed in the cultured neurons is not an artefact of the very high $A\beta$ concentration either more physiological dose of recombinant $A\beta$ or naturally secreted $A\beta$ in conditioned medium should be used.”

We agree that all experiments performed in primary neurons have to be interpreted with great care and cannot fully recapitulate the situation observed in human patients. We are also aware that the opinion which concentration or preparation of $A\beta$ peptides should be used in such a cell culture experiment differs substantially within the field.

To this end various concentrations of $A\beta$ peptides ranging from 0,5 – 100 μ M have been used to

demonstrate AD-linked phenotypes (e.g. see Cantara et al., 2005; Chen et al., 2006; Chafekar et al., 2008; Sondag et al., 2009; Choi et al., 2013; Lee et al., 2017). The concentration used in our study (10 μ M) is well within the range of the published data and even much higher concentrations have been used. We understand, that this is not a conclusive argument to support the view that the concentration employed in our study is optimal but it illustrates that this questions is controversially discussed in the field and thus difficult to answer.

Part of the referees concern might also be due to the fact that there has been some confusion about the A β peptide used in our study. Unfortunately we had sometimes referred to A β_{1-40} and A β_{1-42} . This referee also refers to this issue in “point 8”. We now clarify that we used A β_{1-40} in our experiments. Of note A β_{1-40} has been found less “toxic” than A β_{1-42} in neuronal cell culture experiments and this we why we had actually chosen A β_{1-40} (see also response to point 8). As can be seen from the image to the left, when we use a 10 μ M concentration of A β_{1-42} on hippocampal neurons we observed a very significant detrimental effect in the MTT assay (when compared to A β_{1-40} , see Fig 5) and concluded that in this experimental setting it would be difficult to decipher the impact of anle 138b on A β induced membrane disintegrity vs. other toxic effects.

*Figure: A β_{1-42} oligomers impair cell viability in hippocampal neurons. Hippocampal neurons (DIV 10) were treated with anle138b (1 μ M, anle) or vehicle (veh) before A β_{1-42} oligomers were added (10 μ M, Olig, n=4/group). After 48 h cell viability was measured using the MTT assay. One-way ANOVA revealed a significant group difference. A β_{1-42} oligomers significantly impaired cell viability (t test, *P < 0,0001 vs. control). Anle138b was able to ameliorate this phenotype. Error bars indicate SEM.*

was measured using the MTT assay. One-way ANOVA revealed a significant group difference. A β_{1-42} oligomers significantly impaired cell viability (t test, *P < 0,0001 vs. control). Anle138b was able to ameliorate this phenotype. Error bars indicate SEM.

We like to reiterate that aim of the experiments in neuronal cells was to support the *in vitro* data on A β pores. Thus, we had to chose a concentration in which the employed A β preparation would have only a modest impact on cellular viability as measured in the MTT assays but would impact on membrane integrity. Only in this context we could then test the effect anle 138b in the CyQuant assay to support the hypothesis that anle 138b – at least in part – mediates its therapeutic effect by regulating membrane integrity and A β pores. See page 10, lines 14-19 of the revised manuscript.

However, we take this referees remark very serious and reasoned that in the context of our study it would not help to repeat the experiment using varying concentrations and A β preparations that had been used in the literature so far. We suggest two alternative approaches. First we employed the experimental system show in Fig 5C in a reverse order. To this end we first exposed the cells to A β_{1-40} before anle 138b was added. Our results indicate that in this scenario a significant impairment in cell viability was observed in the MTT assay, that was partially rescued by anle 138b treatment, while membrane integrity measured in the CyQuant assay was completely restored. This data – at least in part - recapitulates the post-plaque group results and clearly shows that prolonged exposure to A β_{1-40} has detrimental effect on cell viability that cannot be fully rescued if anle 138b is added after the onset of pathological changes. However, membrane integrity is fully restored further supporting the view that one important mechanisms by which anle 138b attenuates A β pathology is to protect membranes from the detrimental effects, including that of A β pores.

In addition we have employed the cytochrome release assay (CRA) on isolated mitochondria in the presence of a-synuclein, tau or A β oligomer with or without anle 138b. Our data reveals a substantial damage to mitochoandrial membranes in all conditions that is attenuated by anle 138b (Fig. S7B) suggesting that anle 138b has general effect on membrane integrity.

This data is now show as novel Fig S7B and discussed in the text of the revised manuscript on page 10, lines 15-332.

Referee #2, point 9:

“Please provide details of the preparation and validation of $A\beta^{42}$ monomers and oligomers used. In the method the authors state that these were $A\beta^{40}$ oligomers while in the figure legend and the text that $A\beta^{42}$ was used. This information needs to be clarified.”

Again, we are very thankful that referee 2 points to this discrepancy. We now changed this mistake and clearly state throughout the text that we employed $A\beta_{1-40}$. We also describe this issue in greater detail in the methods section.

This mistake may also help to address the concerns raised by this referee regarding the concentration of $A\beta$ since – although $A\beta_{1-40}$ and $A\beta_{1-42}$ are toxic to neurons - $A\beta_{1-40}$ is considered to be less neurotoxic and was thus employed for cell culture experiments.

We also describe in greater detail the procedure of $A\beta$ preparation. Please see page 20, lines 13-22 of the revised manuscript.

Referee #2, point 10:

“The authors demonstrate that the membrane integrity is better preserved in the anle138b pre-treated neurons when compared to the vehicle pre-treated cells, suggestive that anle138b may prevent formation of the $A\beta$ pores rather than alterations in their kinetics. Reverse treatment, i.e. pre-treatment with $A\beta$ followed by addition of anle138b, should be performed to determine if anle138b affects the pore kinetics.”

We performed the requested experiment that is now presented as novel Fig S7.

Also in the reverse experiment anle 138b treatment ameliorates the $A\beta$ induced loss of membrane integrity, suggesting that anle 138b can also act on already existing pores.

The data is discussed on page 10, lines 15-32 of the revised manuscript. See also response to point 8, referee 2)

Referee #2 points to some minor issues:

“1. Only figure 1 has a number included in the image.”

To facilitate the reading our manuscript for reviewing we have now included the figure number to all images.

“2. The authors describe hippocampal gene expression data on page 6, last paragraph, and refer to figure S1. However, this figure presents data in flies.”

This was indeed a typo and has been corrected. We aimed to refer to figure EV Fig. 1.

“3. Using consistent formatting (colors and decimal places) for different sections in the supplementary table for the pre- and post-plaque group would be helpful.”

We have followed the request of the reviewer.

“4. Please rewrite the section on the formation/dynamic/conductance of the $A\beta$ pores in the lipid bilayers (page 8, second paragraph), as the current version is somewhat confusing.”

We thank the reviewer for this advice and rewrote the text accordingly.

“5. Supplementary figure 3 contains data on the DOPC/DPPC lipid bilayers, which were not used through the manuscript and are not described in the methods. Please refer to the data, explaining why they are relevant or remove it from the manuscript.”

We agree with the reviewer. This data is not relevant for the manuscript since we did not employ DOPC/DPPC bilayers. We have now removed the corresponding panels in Fig S6 (former Fig S3) and instead – in response to point 5 raised by this reviewer – added instead data showing that A β does not alter the thickness of the employed bilayer..

2nd Editorial Decision

15 September 2017

Thank you for the submission of your revised manuscript to EMBO Molecular Medicine and for your note of today. I am happy to say that we have now received the enclosed reports from the referees who were asked to re-assess it. As you will see the reviewers are now supportive and I am pleased to inform you that we will be able to accept your manuscript pending the following final amendments:

1) Please address the text modifications as suggested by referee 1. We do agree with the warnings and maybe tuning down the major clinical claim is desirable. Please provide a letter INCLUDING the reviewer's reports and your detailed responses to their comments (as Word file).

***** Reviewer's comments *****

Referee #1 (Comments on Novelty/Model System for Author):

Authors make some strong statements on the potential human benefit of this compound, even if they use a mouse model for Alzheimer's that only reproduces one aspect of the human form. While this is common practice in the field (all AD mouse models are bad, very limited, reproducing amyloid toxicity, amyloid accumulation, inflammation but not the whole manifestation of the human situation) authors should tone down their conclusion that the results presented here should encourage clinicians to test this compound in clinical trials. That is too far fetched, too premature.

Referee #1 (Remarks for Author):

The authors have satisfactorily addressed my original concerns, both in writing/explanations and with new experiments. Although I do now recommend acceptance, I would like authors (and the editor) to reconsider the final sentence in the Abstract ("In conclusion, our data suggest that anle138b is a novel and promising compound to treat AD-related pathology that should be tested in clinical trials"). First, the key mechanistic aspect of this work is based on experiments performed in artificial membranes, not even in mammalian neurons in vitro. Secondly, the animal model utilised does not reproduce human Alzheimer's disease, only one of the multiple components of this disease (exaggerated amyloid production and deposition). Authors' recommendation to move the analysis of the effect of this compound to clinical trials is premature, unjustified by the data. The literature is full of papers reporting treatments for the "mouse" AD. I am not implying that this compound will also fail in human trials, simply that the current data needs further study and validation before this compound can be considered for clinical trials.

Referee #2 (Remarks for Author):

I have now carefully went through all the author's responses and corrections, and was happy to see that they corrected all the errors and mistakes in labeling that resulted in confusion and difficulty in interpreting their data.

The authors also addressed pretty much all our concerns by providing additional data. Thus, I believe the manuscript is now significantly improved and is suited for publication in the EMBO MM.

1. Please address the text modifications as suggested by referee.

Referee # 1 says “ *The authors have satisfactorily addressed my original concerns, both in writing/explanations and with new experiments. Although I do not recommend acceptance, I would like authors (and the editor) to reconsider the final sentence in the Abstract (“In conclusion, our data suggest that anle138b is a novel and promising compound to treat AD-related pathology that should be tested in clinical trials”). First, the key mechanistic aspect of this work is based on experiments performed in artificial membranes, not even in mammalian neurons in vitro. Secondly, the animal model utilised does not reproduce human Alzheimer’s disease, only one of the multiple components of this disease (exaggerated amyloid production and deposition). Authors’ recommendation to move the analysis of the effect of this compound to clinical trials is premature, unjustified by the data. The literature is full of papers reporting treatments for the “mouse” AD. I am not implying that this compound will also fail in human trials, simply that the current data needs further study and validation before this compound can be considered for clinical trials.*”

We agree of course with the fact that a mouse model for amyloid pathology does not fully recapitulate Alzheimer’s disease and refer to this issue multiple times in our manuscript, e.g. see page 3, lines 28-31, page 10, lines 18-20 or page 13, lines 10-15 etc. of the former revised manuscript.

With respect to translating our findings into the clinics, we like to state that based on the current data we were able to secure funds for a toxicity study. To test anle138b in patients is our ultimate aim. We agree however, that the statement this referee refers to was too strong and have thus changed the last sentence of the abstract. It now reads “In conclusion, our data suggest that anle138b is a novel and promising compound to treat AD-related pathology that should be investigated further.”

Corresponding Author Name: Andre Fischer

EMM-2017-07825-V2